# Adrenal tropism of SARS-CoV-2 and adrenal findings in a post-mortem case series of patients with severe fatal COVID-19

Tanja Paul[1,10], Stephan Ledderose[1,10], Harald Bartsch[1], Na Sun[2], Sarah Soliman[1], Bruno Märkl[3], Viktoria Ruf[4], Jochen Herms [4], Marcel Stern [5], Oliver T. Keppler [5], Claire Delbridge[6], Susanna Müller[1], Guido Piontek[1], Yuki Schneider Kimoto[1], Franziska Schreiber[1], Tracy Ann Williams[7], Jens Neumann[1], Thomas Knösel[1], Heiko Schulz[1], Ria Spallek [8], Matthias Graw[9], Thomas Kirchner[1], Axel Walch[2] & Martina Rudelius [1✉]

Progressive respiratory failure and hyperinflammatory response is the primary cause of death in the coronavirus disease 2019 (COVID-19) pandemic. Despite mounting evidence of disruption of the hypothalamus-pituitary-adrenal axis in COVID-19, relatively little is known about the tropism of severe acute respiratory syndrome coronavirus 2 (SARS-CoV-2) to adrenal glands and associated changes. Here we demonstrate adrenal viral tropism and replication in COVID-19 patients. Adrenal glands showed inflammation accompanied by inflammatory cell death. Histopathologic analysis revealed widespread microthrombosis and severe adrenal injury. In addition, activation of the glycerophospholipid metabolism and reduction of cortisone intensities were characteristic for COVID-19 specimens. In conclusion, our autopsy series suggests that SARS-CoV-2 facilitates the induction of adrenalitis. Given the central role of adrenal glands in immunoregulation and taking into account the significant adrenal injury observed, monitoring of developing adrenal insufficiency might be essential in acute SARS-CoV-2 infection and during recovery.

[1] Institute of Pathology, Faculty of Medicine, Ludwig-Maximilians University Munich, Munich, Germany. [2] Research Unit Analytical Pathology, German Research Center for Environmental Health, Helmholtz Zentrum München, Munich, Germany. [3] Institute of Pathology, University of Augsburg, Augsburg, Germany. [4] Institute of Neuropathology, Ludwig-Maximilians University Munich, Munich, Germany. [5] Max von Pettenkofer Institut, Ludwig-Maximilians University Munich, Munich, Germany. [6] Institute of Pathology, Division of Neuropathology, TUM School of Medicine, Technical University Munich, Munich, Germany. [7] Medizinische Klinik und Poliklinik IV, Klinikum der University Munich, Munich, Germany. [8] Medizinische Klinik und Poliklinik III, Technical University Munich, Munich, Germany. [9] Institute of Forensic Medicine, Ludwig-Maximilians University Munich, Munich, Germany. [10] These authors contributed equally: Tanja Paul, Stephan Ledderose. ✉email: Martina.Rudelius@med.uni-muenchen.de

Coronavirus disease (COVID-19), caused by SARS-CoV-2 has reached a pandemic level with a high global mortality rate. Although SARS-CoV-2 primarily targets the respiratory system, multiple other organs including the kidney, the cardiovascular, and central nervous system can be affected[1–3]. In severe cases of COVID-19, systemic hyper inflammation with the release of cytokines (cytokine storm) is difficult to manage and can result in multiple organ failure[4]. An intact hormone balance, especially of adrenal hormones such as cortisol, is critical to master this challenge. Cytokine toxicity and autoimmunity can be reduced by mild immunosuppression due to activation of the hypothalamus-pituitary-adrenal axis (HPA axis) with hypercortisolemia[5]. However, disruption of the HPA axis can cause dysregulation of the physiological innate immune response against viral infection[6]. For the SARS-CoV 2002–2004 outbreak, it has been shown that SARS-CoV can directly injure adrenal glands[7].

Only a few reports of adrenal findings of COVID-19 patients have been published recently. Two autopsy series showed varying frequencies of adrenal damage[8,9]. Two case reports and one clinical series that evaluated CT findings could link adrenal infarction or hemorrhage to adrenal insufficiency in COVID-19 patients[10–12]. However, detailed morphologic and molecular analysis of adrenal glands with simultaneous evidence of viral adrenal tropism for SARS-CoV-2 are lacking to date.

Here, we show adrenal tropism of SARS-CoV-2 associated with local inflammation and severe adrenal damage in COVID-19 patients. In addition, COVID-19 specimens were characterized by a specific metabolic profile with significantly reduced Cortisone intensities.

## Results

**Adrenal tropism of SARS-CoV-2.** As ACE2 and TMPRSS2 facilitate SARS-CoV-2 infection we performed immunohistochemistry for ACE2 and TMPRSS2. ACE2 was expressed moderately in adrenal cortical cells and strongly in intervening capillaries (Fig. 1A). TMPRSS2 immunohistochemistry was positive in the cytoplasm of adrenal cells with a few cells with nuclear staining (Fig. 1B). This distribution was confirmed by double immunofluorescence with CD34 staining capillaries and SF1 staining adrenal cells (Fig. 1D–I). ACE2 colocalized with CD34 (Fig. 1D) (95 ± 3% of CD34 positive cells were double positive for CD34 and ACE2) and SF1 (Fig. 1G) (87 ± 4% of SF1 positive cells were double positive for SF1 and ACE2). TMPRSS2 colocalized with SF1 (Fig. 1H) (84 ± 4% of SF1 positive cells were double positive for SF1 and TMPRSS2), however, no colocalization with CD34 could be observed (Fig. 1E) (0% of CD34 positive cells were double positive for CD34 and TMPRSS2).

We could detect s-SARS-CoV-2 protein by immunohistochemistry in adrenal cortical cells of 19/19 patients with positive postmortem SARS-CoV-2 PCR (Fig. 1C). In double immunofluorescence images, s-SARS-CoV-2 colocalized with SF1 (Fig. 1I) (32 ± 11% SF1 positive cells were positive for s-SARS-CoV-2), however, s-SARS-CoV-2 did not colocalize with CD34 positive cells (Fig. 1F) (0% of CD34 cells were double positive for s-SARS-CoV-2). Viral tropism to adrenal cells was further validated by in situ hybridization. Sense s-SARS-CoV-2 RNA could be clearly visualized in the adrenal cortex (Fig. 1J, K). Additional detection of antisense s-SARS-CoV-2 RNA confirmed viral replication in the cells (Fig. 1L, M). Viral RNA was detected in SF1 positive adrenal cells and in CD34 positive endothelial cells as shown by multiplex immunofluorescence and in situ hybridization (Fig. 1N, O).

Expression of ACE2, TMPRSS2, and SARS-CoV-2 in adrenal tissue was confirmed by western blot analysis with variable amounts of ACE2, TMPRSS2, and s-SARS-CoV-2 protein detectable (Fig. 1O).

**Viral infection and replication in adrenal cortical carcinoma cell line.** To functionally validate our observations of autopsy tissue of COVID-19 patients we performed infection experiments with two adrenal cortical cell lines (HAC15 which was positive for ACE2 and TMPRSS2 and SW13 which showed weak ACE2 expression and was negative for TMPRSS2) (Fig. 2A). Target cells were treated with either RDV (1 μM) 2 h before infection or were left untreated. Subsequently, cells were challenged with a serial dilution of a stock of an expanded SARS-CoV-2 clinical isolate. Viral replication of SW13 and HAC15 was quantified by RT-qPCR, both in a wash (postwash; time zero) and in the supernatant (harvest; time 72 h). Additionally, cells were collected for histopathology and western blot analysis. SW13 showed no detectable virus replication with equal ct values of postwash and harvest, however, HAC15 showed low virus replication with ct values of harvest clearly exceeding the postwash values and reaching almost the inoculum (Fig. 2B). Virus replication could be inhibited by RDV pretreatment. FISH analysis of cell blocks and western blot analysis confirmed RT-qPCR results. In RDV treated HAC15 no sense or antisense SARS-CoV-2 RNA could be detected (Fig. 2C), in contrast, untreated HAC15 showed virus infection and replication with positivity for antisense and sense SARS-CoV-2 (Fig. 2C) and expression of SARS-CoV-2 protein (Fig. 2D). Intriguingly, low viral replication was sufficient to induce apoptosis in HAC15 with cleavage of caspase 3 shown by western blot analysis (Fig. 2D) and by immunohistochemistry (Fig. 2C). Moreover, SARS-CoV-2 triggered inflammatory response with upregulation of IL-6 (Fig. 3D), which is elevated in COVID-19 patients and correlates with adverse clinical outcome[13,14].

**Inflammation of the adrenal gland.** In all adrenal glands of SARS-CoV-2 positive patients lymphocytic and histiocytic, infiltrates were detected, H&E images of each patient ($n = 31$) is given in supplemental Fig. 3. Infiltrates often formed small foci consistent with 50 or more lymphocytes and histiocytes. Foci were accentuated perivascular and were observed in the medulla as well as in the adrenal cortex (zona fasciculata, Fig. 3A). It is known that lymphocytic infiltrates can be found in the medulla of normal adrenal glands and are not indicative of adrenal inflammation, therefore we focused our analysis on the adrenal cortex. Multiplex immunohistochemistry (Fig. 3D–G) showed that in the cortex of COVID-19 patients CD4 positive T-cells (Fig. 3D, H) were more numerous than in influenza patients (mean number/HPF 17.5 ± 3.3 versus 9.2 ± 1.9; $p < 0.0001$) and CD8 positive T-cells (Fig. 3E, I) were more numerous in COVID-19 patients as well (mean number/HPF 6.3 ± 1.6 versus 2.0 ± 1; $p < 0.0001$). With immunohistochemistry of cleaved caspase 3 (Fig. 3C) we were able to detect apoptosis in the center of 79% of the inflammatory infiltrates (mean = 3 ± 3). In multiplex immunofluorescence significantly more CD68 positive macrophages (Fig. 3F, J) were found in the cortex of COVID-19 patients compared to influenza patients (mean number/HPF 27.6 ± 9 versus 16.7 ± 3.7; $p < 0.0001$).

**Adrenal thrombosis and hemorrhage.** We examined the adrenal vasculature with H&E stain, Masson´s trichrome stain, and anti-fibrin immunohistochemistry. We did not observe vascular endothelialitis (as described in the lung or renal specimens of COVID-19 patients[2]). Furthermore, no thrombi were detected in precapillary vessels. However, in 11/19 (58%) of COVID-19 patients capillaries were expanded and microthrombi were clearly visible (Fig. 4A, B) as demonstrated by anti-fibrin immunohistochemistry. Adrenal capillary microthrombi were three times as prevalent as in patients with influenza (mean ± SD number of distinct thrombi per square centimeter of vascular lumen area, 45.1 ± 2.9 and 15 ± 1.8; $p < 0.0001$). In three COVID-19 patients

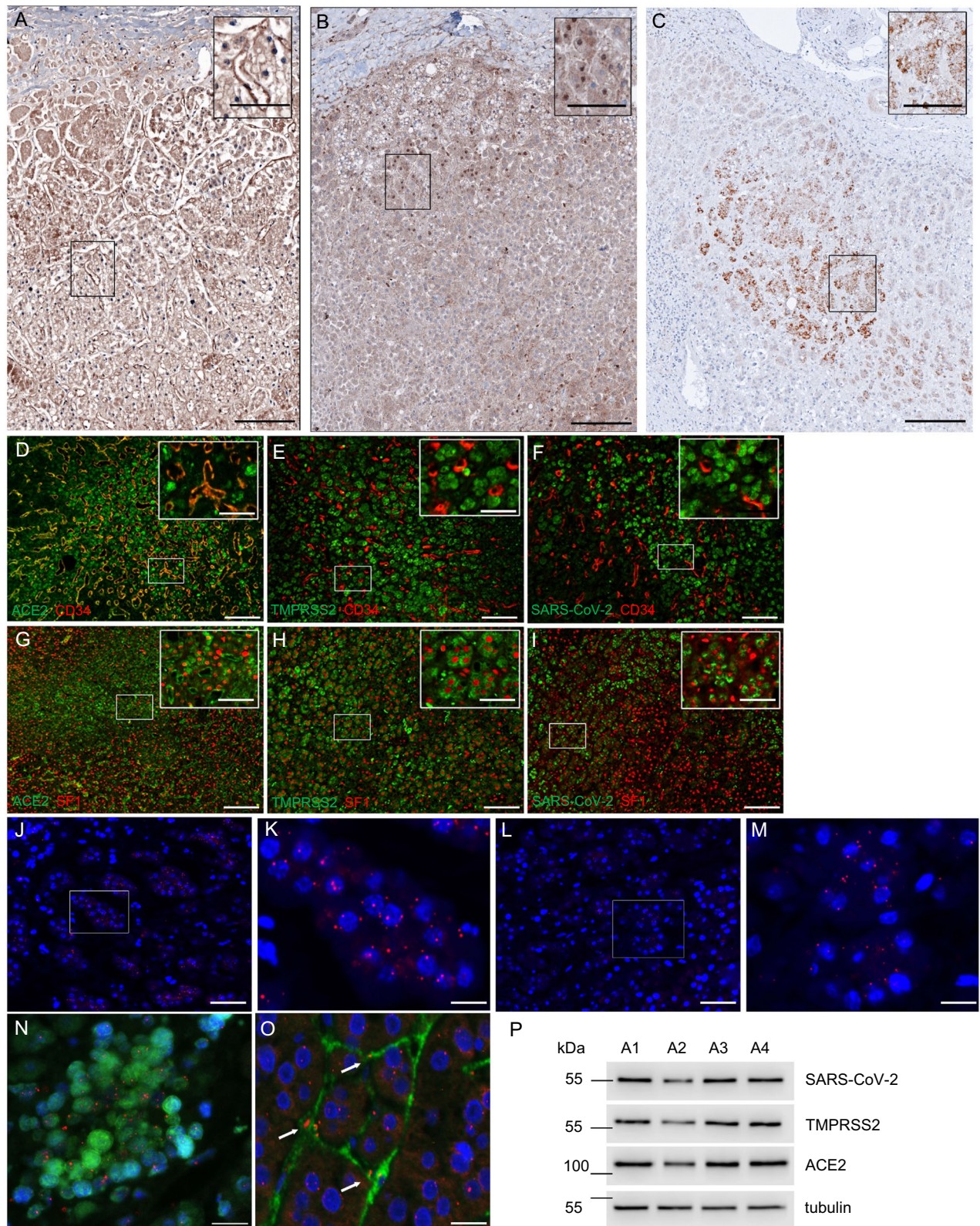

(16%) focal fresh hemorrhage and ischemic infarction could be detected in the adrenal cortex or medulla (Fig. 4C).

**Disruption of the adrenal cortex zonation.** The normal adrenal cortex is structured in three zones (zona glomerulosa, zona fasciculata, and zona reticularis), which can be easily identified by H&E at low magnification and can be highlighted by immunohistochemistry with CYB11B2 (zona glomerulosa) and CYB5A (zona reticularis) (Fig. 5A). All adrenal glands from the SARS-CoV-2 positive group revealed moderate to severe disruption of the zonation (Fig. 4B, C) with 47% ($n = 9$) of disruption grade 2 and 53% ($n = 10$) of disruption grade 1. In contrast in the influenza group 30% ($n = 3$) showed no disruption, 40% ($n = 4$)

**Fig. 1 Adrenal tropism of SARS-CoV-2.** Overview images of adrenal glands of COVID-19 patients ($n = 20$) (scale bar $= 100\,\mu m$) with magnified inlets (scale bar $= 15\,\mu m$) of immunohistochemistry of ACE2 (**A**), TMPRSS2 (**B**), and s-SARS-CoV-2 (**C**) are shown. Adrenal cells and intervening capillaries express ACE2 and TMPRSS2 and s-SARS-CoV-2 protein can be detected in adrenal cortical cells. Double immunofluorescence of adrenal glands of COVID-19 patients ($n = 19$) of ACE2 (green) and CD34 (red) (**D**), TMPRSS2 (green) and CD34 (red) (**E**), s-SARS-CoV-2 (green) and CD34 (red) (**F**), ACE2 (green) and SF1 (red) (**G**), TMPRSS2 (green) and SF1 (red) (**H**), and s-SARS-CoV-2 (green) and SF1 (red) (**I**) are displayed as overview (scale bar $= 100\,\mu m$) with magnified inlets (scale bar corresponds $= 15\,\mu m$). ACE2 colocalizes with SF1 and CD34, whereas TMPRSS2 and SARS-CoV-2 only localize to SF1 but not to CD34. In-situ hybridization of adrenal tissue with s-SARS-CoV-2 antisense probe (**J**, **K** in red) and sense probe (**L**, **M** in red) with DAPI counterstain (blue) (**J**, **L** (overview (scale bar $= 50\,\mu m$) and corresponding magnification (**K**, **M**) (scale bar $= 15\,\mu m$) confirm the presence of s-SARS-CoV-2 RNA in adrenal tissue ($n = 19$). Viral RNA (red dots) is detected in SF1 positive adrenal cells (green, **N**) and in CD34 positive endothelial cells (green, **O**); scale bar corresponds to $10\,\mu m$, DAPI counterstaining in blue. Western blot analysis of adrenal tissue ($n = 4$) (**P**) shows a variable expression of ACE2, TMPRSS2, and s-SARS-CoV-2 in adrenal tissue (tubulin is shown as loading control).

disruption grade 1 and 30% ($n = 3$) disruption grade 2. Compared to the matched group of influenza patients disruption of the zonation was significantly more pronounced in the COVID-19 group ($p = 0.041$) (Fig. 5E).

Interestingly all ten COVID-19 patients with no and one patient with only short ICU stay (≤7 days) also showed moderate to severe disruption of adrenal cortex zonation. Two COVID-19 patients with seroconversion and extended ICU stay were characterized by massive interstitial fibrosis with collagen matrix production (Fig. 5D) and severe disruption of the architecture, whereas lymphocytic infiltrates could not be detected.

**Hypothalamus-pituitary axis.** There is currently no evidence of direct hypothalamic or pituitary effects of SARS-CoV-2 infection. We conducted a histomorphologic analysis of the hypothalami and the pituitary glands of ten COVID-19 patients. Morphologically and immunohistochemically no significant inflammatory infiltrate was detected. The architecture was unremarkable without evidence of hemorrhage or necrosis (Fig. 5F, G). In summary, no significant structural abnormalities could be identified in postmortem specimens from COVID-19 patients.

**Metabolomic alterations of adrenal glands associated with COVID-19.** For spatial metabolomics MALDI-FT-ICR imaging mass spectrometry analysis was conducted, comparing four FFPE specimens of adrenal glands of COVID-19 patients with four adrenal glands of influenza patients (specimen with no signs of autolysis were chosen). Overall, 4663 MS peaks within the mass range of m/z 75 to 1000 could be resolved within the tissue (Fig. 6B).

Metabolic data were exported and used for hierarchical clustering and component analyses. In hierarchical cluster analysis, the two groups (COVID-19 and influenza) could be clearly distinguished based on the metabolite patterns, as demonstrated by heat map analysis (Fig. 6A). The distribution of metabolites revealed obvious differences between COVID-19 and influenza (Fig. 6D). Especially, phosphatidic acid and phosphatidylinositol showed high intensity in COVID-19 patients, whereas inositol cyclic phosphate and ribose phosphate showed high intensity in the influenza group.

Pathway enrichment analysis was performed on the two different groups. Enriched metabolic pathways contributing to group discrimination include glycerolphospholipid metabolism, pentose phosphate pathway, and ascorbate and aldarate metabolism (Fig. 6E, F).

Steroid metabolome analysis showed the reduced intensity of cortisone in the COVID-19 group ($p = 0.648$) compared to the influenza group (Fig. 6C).

## Discussion

In this study, we examined the morphologic and molecular features of 21 adrenal glands obtained during autopsy and ten

postmortem specimens of the pituitary gland and the hypothalamus from patients who died from COVID-19. Findings were compared to specimens of patients who died from influenza A (H1N1) infection ($n = 10$). We detected SARS-CoV-2 RNA and protein in adrenal glands associated with substantial inflammation and injury. HAC15 a cortical adrenal carcinoma cell line showed a low level of viral replication when infected with SARS-CoV-2, however, this was sufficient to induce apoptosis and inflammatory response. Distinctive pathologic features of adrenal specimens from COVID-19 patients were lymphohistiocytic infiltrates, widespread micro thrombosis, and disruption or loss of adrenal cortical zonation. Metabolomic imaging could clearly separate the specimens of each group by cluster analysis. Enriched metabolic pathways contributing to group discrimination include glycerolphospholipid metabolism, pentose phosphate pathway, and ascorbate and aldarate metabolism, moreover, specimens of COVID-19 patients showed a substantial reduction in cortisone levels.

ACE2 and TMPRSS2, which have been shown to contribute to viral entry into the cell and to viral persistence[15] were expressed in adrenal and endothelial cells as shown previously for ACE2[16]. SARS-CoV-2 was detected by immunohistochemistry or in situ hybridization in all adrenal glands of patients with positive postmortem nasopharyngeal swabs. S-SARS-CoV-2 protein and RNA could be mapped to adrenal cells. To our knowledge, this is one of the most comprehensive reports of adrenal findings of patients who died from COVID-19, which indicates adrenal tropism of the virus. Although the productive viral infection was not demonstrated, detection of antisense SARS-CoV-2 RNA in adrenal cells confirmed viral replication in adrenal cells. Viral RNA and protein was imaged with high spatial resolution and viral RNA could be detected not only in adrenal cells but also in endothelial cells of intervening capillaries. As SARS-CoV-2 protein and antisense RNA was not observed in endothelial cells detection of SARS-Cov-2 sense RNA in endothelial cells might rather be indicative of viral transmission from the bloodstream than indicative for viral replication. In infection experiments with an adrenal cortical carcinoma cell line (HAC15), low-level viral replication was observed. Even though these are preliminary data from a carcinoma cell line, infection-induced apoptosis and triggered inflammatory response with enhanced expression of IL-6, which is elevated in COVID-19 patients and correlates with adverse clinical outcome[17,18].

We observed inflammation of the adrenal glands with perivascular accentuation as well as ongoing apoptosis in the center of the infiltrates. Quantification of T-cell subsets and macrophages confirmed a significant increase of inflammatory infiltrates in COVID-19 patients compared to influenza patients. This is in line with previous studies[8,9,19] reporting variable frequencies of adrenal inflammation, however to date comparison to other postmortem adrenal gland specimens (such as Influenza) was missing. Our findings of viral identification in the adrenal gland

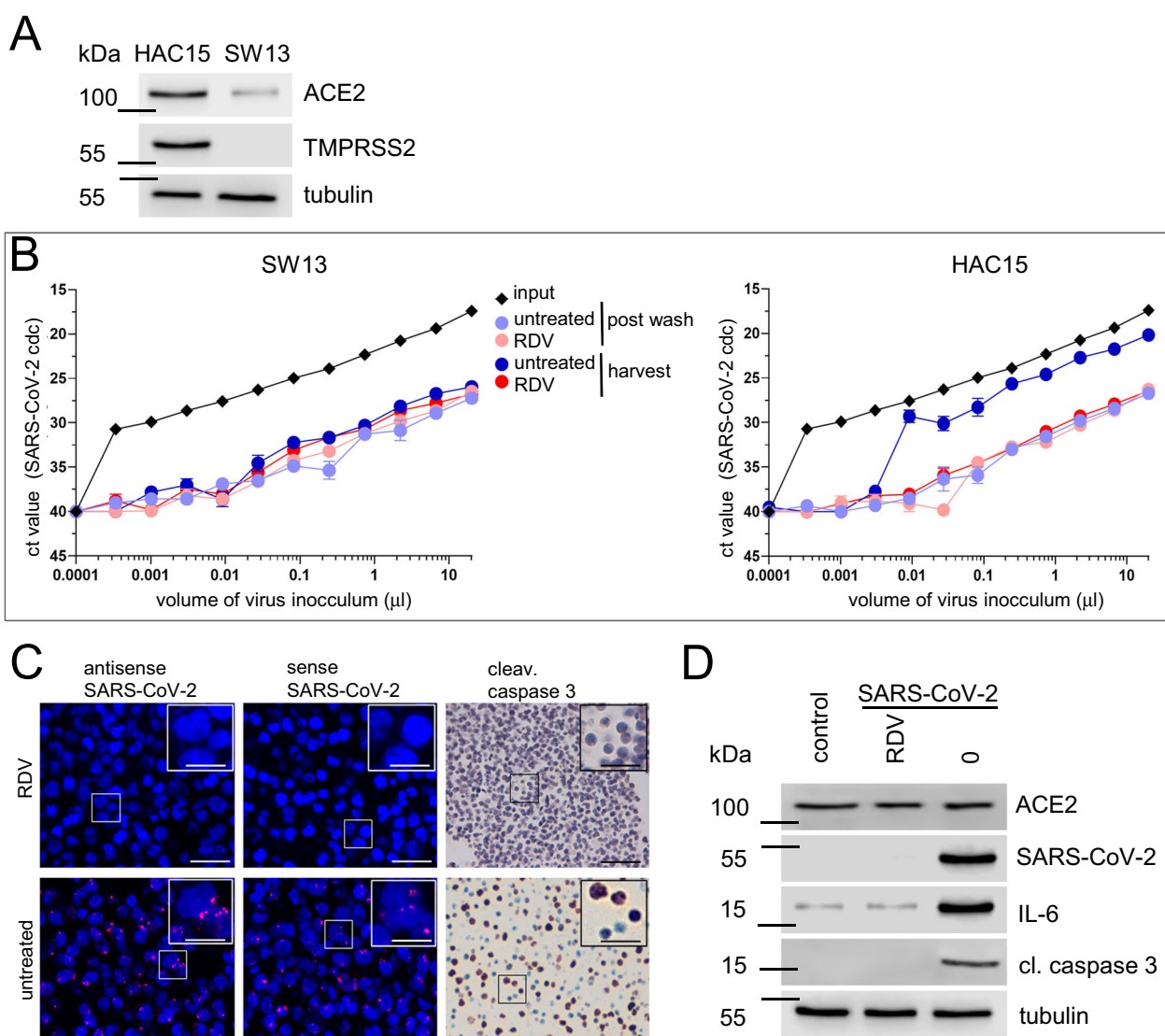

**Fig. 2 Infection of adrenal cortical carcinoma cell lines with SARS-CoV-2 induces apoptosis and triggers an inflammation response. A** Western blot analysis of lysates from SW13 and HAC15 cells (three biological replicates) for expression of ACE2 and TMPRSS2, tubulin served as a loading control. **B** SW13 and HAC15 cells were either pretreated with RDV (red) or left untreated (blue) and challenged with the indicated volume of an expanded clinical SARS-CoV-2 isolate (input). The highest inoculum corresponds to a multiplicity of infection (MOI) of 0.05. Supernatant aliquots were taken postwash on day 0 to determine the background of the residual inoculum in the absence of replication. After 3 days in culture, additional supernatant aliquots were taken and analyzed. Shown are RT-qPCR analyses for the SARS-CoV-2 viral load as assessed by the Ct value for the SARS-CoV-2 *N1* gene. Depicted are the mean and standard deviation of two technical replicates from three independent experiments. Source data are provided as a source data file. **C** In situ hybridization and immunohistochemistry of infected HAC15 cells (MOI of 0.03), either pretreated with RDV (RDV+) or left untreated (RDV−). Two technical replicates from three independent experiments were completed. Shown are overview images (scale bar = 50 μm) and magnified insets (scale bar = 10 μm)). Shown are antisense and sense RNA to detect SARS-coV-2 RNA (red dots; left and middle panel) and protein of cleaved caspase 3 (right panel. Nuclear counterstain (DAPI, hematoxylin) is shown in blue. **D** Western blot analysis of lysates of HAC15 cells (control, infected (untreated (0), and infected (pretreated with RDV (RDV)) for ACE2, SARS-CoV-2 nucleocapsid protein, cleaved caspase 3, and IL-6. Tubulin served as a loading control. Two technical replicates from three independent experiments were completed with an MOI of 0.03.

and accumulation of inflammatory cells accompanied by inflammatory cell death suggest that SARS-CoV-2 facilitates the induction of adrenalitis as a direct consequence of viral involvement.

Hypercoagulability with immunothrombosis and thromboembolic events is a characteristic finding in COVID-19 patients and is associated with an increased risk of death[20]. In more than half of COVID-19 patients (58%) microthrombi in capillaries were present and focal hemorrhage or infarction was detected in 16% of COVID-19 patients. Moderate to severe adrenal injury was present in COVID-19 patients with significant disruption of the adrenal zonation. It can not be entirely ruled out that these changes are due to other nonspecific systemic critical illness[21], however adrenal injury of COVID-19 patients was significantly more severe than in age-matched influenza controls. Moreover, the adrenal injury was not only detected in patients with extended ICU stay but also in patients with no ICU stay. Notably, two patients with prolonged ICU stay and seroconversion showed severe damage with prominent scarring. Adrenal infarction with associated adrenal insufficiency in COVID-19 patients has been described in previous radiological studies and case reports[10–12].

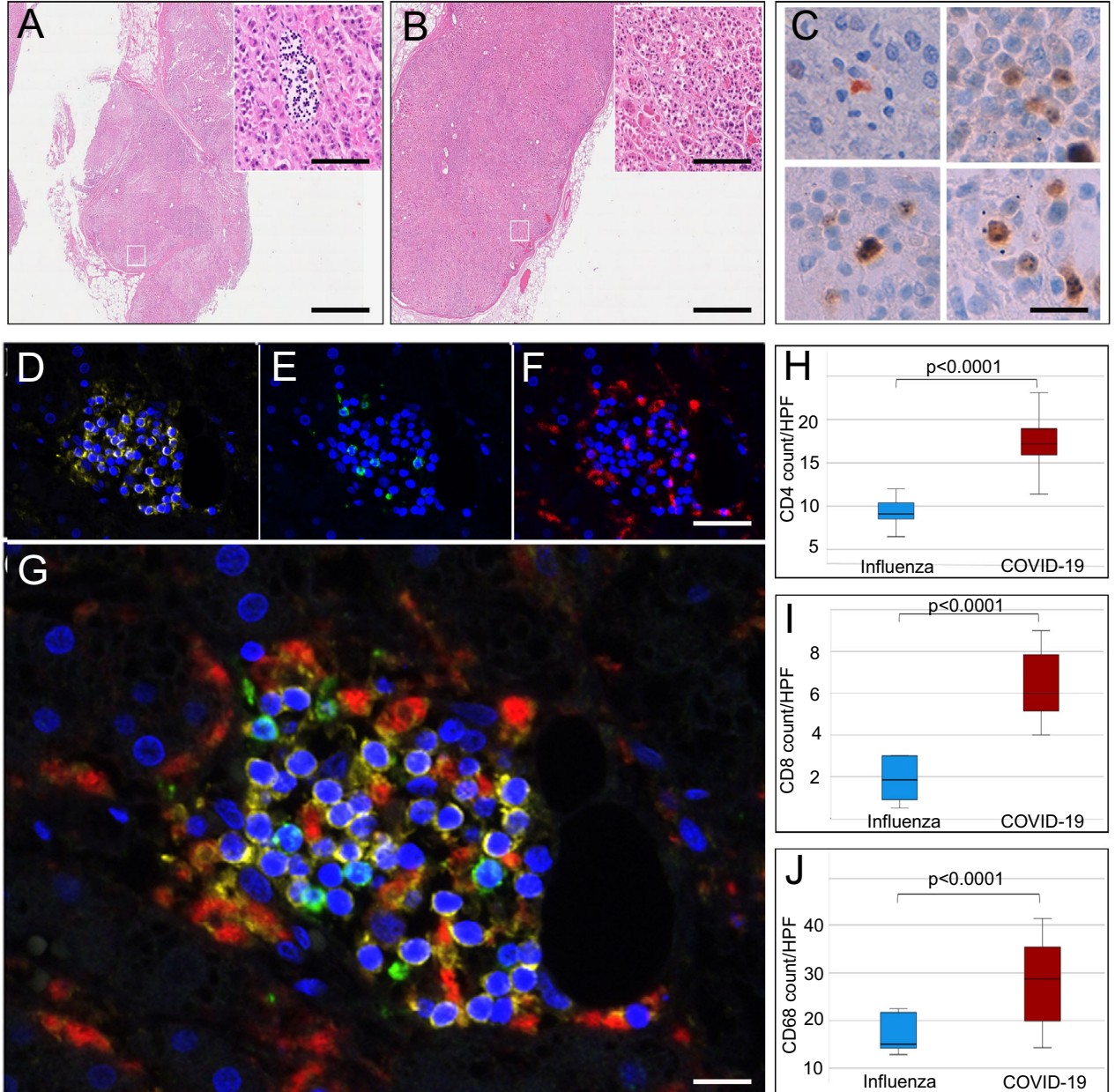

**Fig. 3 Inflammation of adrenal glands from patients who died of COVID-19 or influenza.** H&E sections of adrenal glands of COVID-19 patients (**A**) and influenza patients ($n = 30$) (**B**) (overview scale bar $= 800\,\mu$m and magnified inlet scale bar $= 50\,\mu$m) reveal lymphohistiocytic infiltrate in the adrenal cortex of COVID-19 patients. Multiplex immunofluorescence of adrenal glands ($n = 19$) (**D**–**G**, scale bar $= 50\,\mu$m) shows that infiltrates consisted of CD4 (**D**, yellow), CD8 (**E**, green), and CD68 (**F**, red) positive cells (DAPI counterstain in blue). Box and Whisker plots (**H**, **I**) display lower quartile, upper quartile, and median bounds of cohort expression at the box's minima, maxima, and centerlines, respectively. Whisker lines display lower (bottom) and upper (top) extreme value ranges. $p$ values were calculated by an unpaired two-sided Student's $t$-test. Source data are provided as a Source Data file. Quantification of cell types reveals a significantly higher number ($p < 0.0001$) of CD4 (**H**), CD8 (**I**), and CD68 (**J**) positive cells in patients with COVID-19 compared to influenza. In the center of inflammatory infiltrates ($n = 19$) single-cell apoptosis could be detected by immunohistochemistry of cleaved caspase 3 (**C**, scale bar corresponds to 10 $\mu$m).

Metabolomic imaging of adrenal glands by MALDI could clearly separate COVID-19 patients from influenza patients. COVID-19 patients were characterized by activation of the gly-cerophospholipid metabolism with an increase of specific inter-mediates and metabolites such as phosphatidic acid and phosphatidylinositol. A decrease of serum levels of glycerophos-pholipids has been correlated with a severe clinical course of COVID-19[20] and corresponding metabolites have been found significantly elevated in Coronavirus-infected cells[22]. The role of

glycerophospholipid metabolism in hypoxic stress response[23] and phagocytosis is well documented[24]. In addition, steroid profiling showed a decrease of cortisone intensities, which might indicate disturbed adrenal cortical steroid production.

Adrenal hormones play a crucial role in the immune response, however comprehensive studies investigating the hypothalamus-pituitary-adrenal axis in COVID-19 patients are missing so far. In contrast to our remarkable findings in adrenal glands, we could not observe significant structural abnormalities or inflammatory

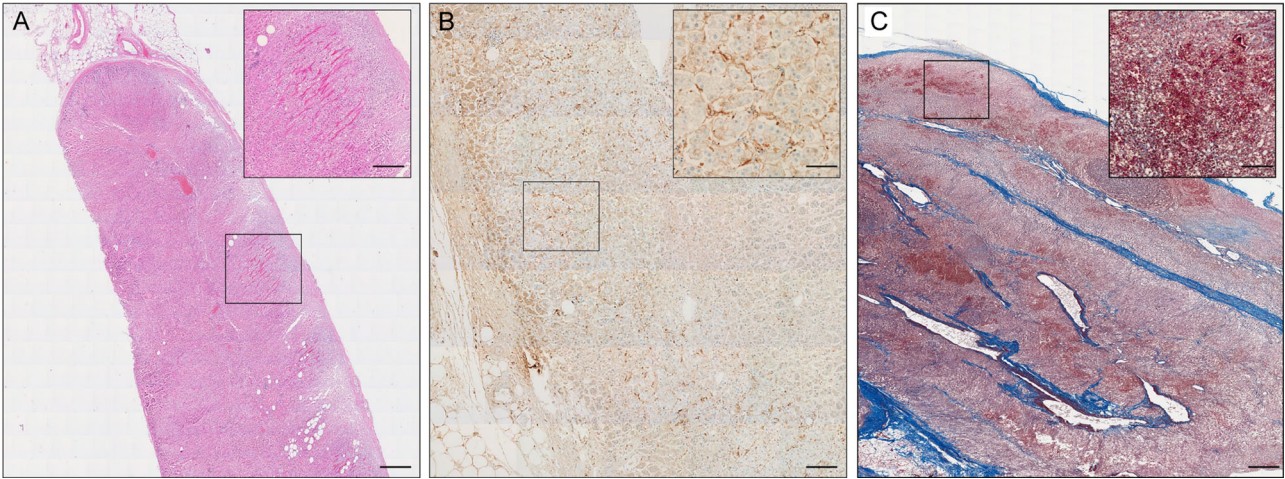

**Fig. 4 Microthrombi and focal hemorrhage in adrenal glands from COVID-19 patients.** Hematoxylin-eosin staining (**A**), and anti-fibrin immunohistochemistry (**B**) (overview (scale bar = 100 μm and magnified inlet (scale bar = 10 μm) (n = 19) shows expanded capillaries with fibrinous microthrombi. In a subset of patients focal fresh hemorrhage could be observed (**C**), Masson's trichrome staining (overview (scale bar = 100 μm and magnified inlet (scale bar = 10 μm).

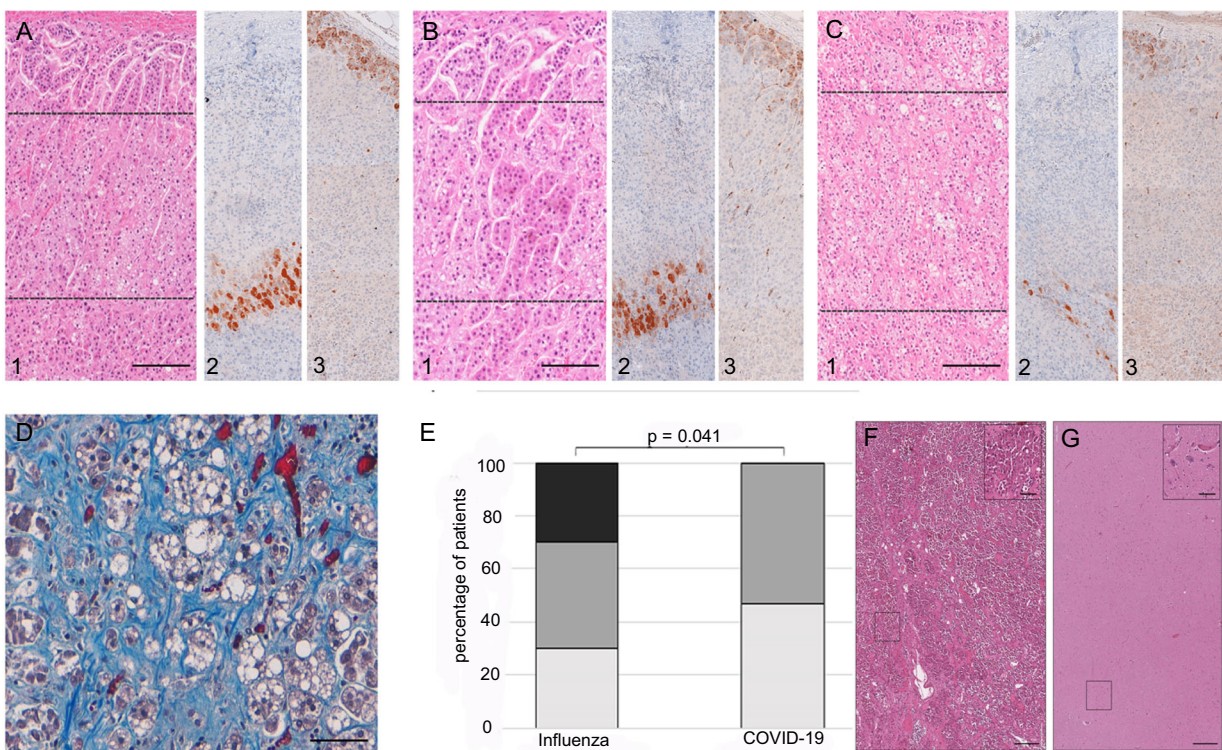

**Fig. 5 Disruption of zonation in adrenal glands from COVID-19 patients.** Histopathological examination of adrenal glands (n = 30) reveals disruption of adrenal gland zonation in COVID-19 patients. In panel **A** normal zonation in zona glomerulosa, zona fasciculata, and zona reticularis is shown with hematoxylin-eosin staining (H&E) (A1) and immunohistochemistry for CYB5A (zona reticularis, A2) and for CYB11B2 (zona reticularis, A3). Panel **B** displays moderate disruption of the zonation and in panel **C** zonation of the cortex is not visible in H&E staining and is severely disrupted as visualized by immunohistochemistry. Quantification of disruption (score 0 (intact zonation, black), score 1 (moderate disruption, gray), and score 2 (severe disruption, white)) and statistical analysis (**E**) showed significantly more severe disruption of zonation in COVID-19 patients compared to influenza patients (p values were calculated using Chi-squared test). Source data are provided as a Source Data file. Marked interstitial fibrosis with collagen matrix production indicated in blue (**D**, Masson's trichrome stain) was noticed in patients with seroconversion. Scale bars in **A**–**C** correspond to 200 μm in **D** to 100 μm. H&E staining of the pituitary gland and hypothalamus (**F**, **G**) shows no significant alterations (overview (scale bar = 100 μm and magnified inlet (scale bar = 10 μm).

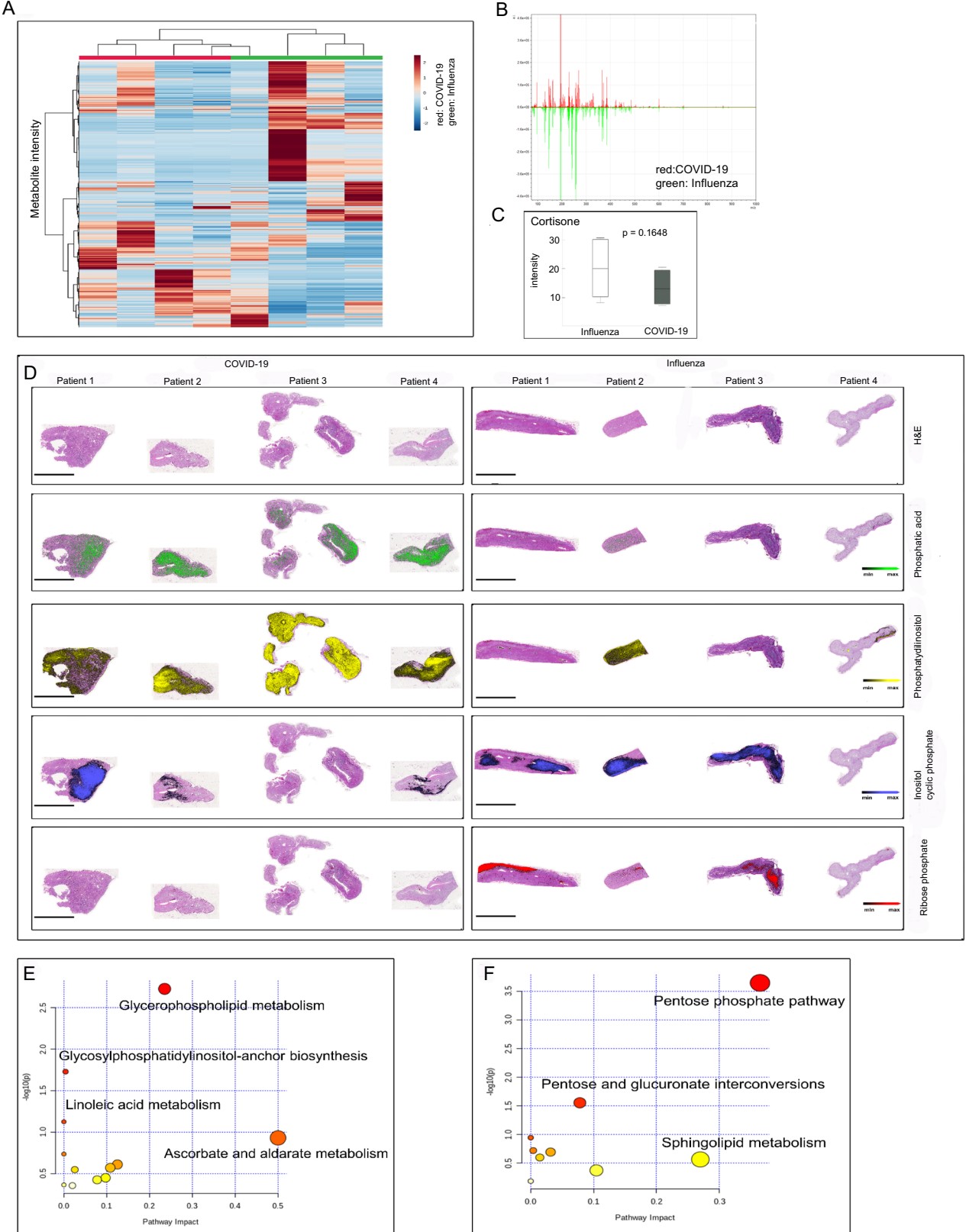

infiltrates in the hypothalamic or pituitary glands of COVID-19 patients. However, the severe damage of adrenal glands might lead to primary adrenal insufficiency with dysregulation of the HPA axis and hypocortisolemia in the end. Adrenal insufficiency is not a rare manifestation in COVID-19. In addition to several case reports[10,25–29] some observational studies[12,17] also reported

this presentation. Like SARS-CoV, SARS-CoV-2 contains various amino acid sequences with homology to human adrenocortico-tropic hormone (ACTH) key residues[18]. Therefore, antibodies against the virus could eventually target the host ACTH and induce a relative cortisol insufficiency. In the acute setting in severely ill COVID-19 patients hypercortisolemia is prevalent[30],

**Fig. 6 Distinct metabolomics phenotype of adrenal glands from COVID-19 patients.** MSI was performed for COVID-19 and Influenza specimens ($n = 8$, **A**–**F**). In heat map-based clustering analysis of identified mass peaks (**B**), COVID-19 patients could be clearly distinguished from influenza patients (**A**). Each colored cell corresponds to an intensity value, with samples in columns and features in rows (**A**). Examples of MSI scans (**D**) show activated glycerophospholipid metabolism in COVID-19 with an accumulation of intermediates and metabolites such as phosphatic acid and phosphatidylinositol. Pathways enriched in the COVID-19 group are glycerophospholipid metabolism and ascorbate and aldarate metabolism (**E**), whereas the influenza group is characterized by a high pentose phosphate pathway (**F**). Steroid profiling shows a reduction in cortisone intensity in adrenal glands from COVID-19 patients (**C**; $p$ value $= 0.1648$). Box and Whisker plots display lower quartile, upper quartile, and median bounds of cohort expression at the box's minima, maxima, and centrelines, respectively. Whisker lines display lower (bottom) and upper (top) extreme value ranges. $p$ values were calculated by an unpaired two-sided Student´s $t$-test. Source data are provided as a Source Data file.

## Table 1 Clinical characteristics of patients.

| Parameter | COVID-19 N (if applicable); % or range | Influenza N (if applicable); % or range |
|---|---|---|
| Sex | | |
| male | 13; 68 % | 8; 80% |
| female | 6; 32% | 2; 20% |
| Age: mean (years; range) | 73; 60–91 | 76; 53–84 |
| Length of ICU stay (days; range) | 9; 0–36 | 8; 0–32 |
| Cause of death | | |
| respiratory failure | 12; 63% | 7; 70% |
| cardiorespiratory failure | 4; 21% | 2; 20% |
| multiorgan failure | 3; 16% | 1; 10% |
| Comorbidities | | |
| cardiovascular disease | 15; 79% | 6; 60% |
| hypertension | 19; 100% | 10; 100% |
| diabetes mellitus | 7; 37% | 5; 50% |
| chronic kidney disease | 2; 11% | 0; 0% |

which might be attributed to reduced serum protein binding, persistent suppression of metabolic degradation, and stress-induced decrease of cortisol clearance[21]. However, preliminary results of the RECOVERY trial (Randomized evaluation of COVID-19 Therapy; ClinicalTrials.gov identifier:NCT04381936) indicate a so-far unexplained survival benefit of low dose usage of steroids in critically ill COVID-19 patients[31]. Moreover, Leow et al.[32] reported hypocortisolism in forty percent of patients three months after recovery from SARS. Moreover fatigue and malaise characteristics for long-COVID-19[33] are common in patients with adrenal insufficiency. Therefore, considering our findings of severe impairment of adrenal glands in COVID-19 patients monitoring for adrenal insufficiency during and after SARS-CoV-2 infection seems essential to avoid adrenal crisis.

In conclusion, we found adrenal tropism of SARS-CoV-2 in critically ill patients accompanied by adrenalitis and pronounced damage of adrenal glands with disturbance of adrenal metabolism. Given the central role of the adrenal glands in stress response and immune regulation this finding is of high clinical relevance and should lead to close monitoring of the hypothalamus-pituitary-adrenal- axis in COVID-19 patients in the acute stress situation and in recovery.

## Methods
**Study design and patient selection**. The study was approved by the ethics committee of the LMU (project number 20-1039), informed consent was obtained from the legal representatives of deceased individuals and the study complies with all relevant ethical regulations.

We analysed adrenal gland autopsy specimens from 21 patients who died from COVID-19. All patients were diagnosed with COVID-19 ante mortem and PCR-tests for SARS-CoV-2 from postmortem nasopharyngeal swabs were positive except for two patients with known seroconversion. In addition, the hypothalami and pituitary glands of ten patients were analyzed. Adrenal glands were compared to those from ten patients who died from pneumonia caused by influenza A virus H1N1. Specimens from patients with influenza were archived tissues and chosen for the best possible match with respect to age, sex, and disease severity. Clinical features are detailed in Table 1. The mean duration of intensive care unit stay (ICU) was comparable between COVID-19 patients and influenza patients; in the COVID-19 group ten patients with no intensive care unit stay (ICU) were included and in the influenza group one patient did not stay on the ICU.

**Histological and immunohistochemical evaluation**. All adrenal glands were comprehensively analyzed by standard histopathology and histochemical methods (Hematoxylin and Eosin (H&E) and Masson´s trichrome stains). Adrenal cortical zonation was scored semiquantitatively based on the identification of the three zones (zona glomerulosa, zona fasciculata, and zona reticularis) in the adrenal cortex at 5 x magnification. Zonation was graded as follows: 0 when the three adrenocortical zones were clearly visible, (1) when zonation was moderately disturbed and (2) when the three different zones were only distinguished by immunohistochemistry.

Standard immunohistochemistry was performed for detection of adrenal cortical zonation (Cytochrome P450 family 11 subfamily B member 1 (CYB11B2) and Cytochrome B5 (CYB5A)), apoptosis (cleaved caspase 3), and expression of angiotensin-converting enzyme 2 (ACE2) and transmembrane protease serine 2 precursor (TMPRSS2). After antigen retrieval slides were incubated with antibodies against CYB11B2 (1:50; Proteintech, United Kingdom), CYB5A (1:120; Atlas, Sweden) cleaved caspase 3 (1: 100; Cell Signaling, USA), ACE2 (1:100; Abcam, Germany), or TMPRSS2 (1:70; Bio SB, USA). Detection was carried out with ImmPress anti-rabbit IgG polymer kit (Vector Laboratories, USA) or MACH 3 Mouse HRP Polymer detection (Biocare Medical, USA), respectively, using the manufacturer´s protocol. For multispectral imaging CD8 (C8/144B, dilution 1:150, Medac GmbH, Germany), CD4 (EP204, 1:200, Medac GmbH), CD34 (QEBnd-10, 1:100, DAKO), steroidogenic factor 1 (SF1) (A-1, 1:75, Santa Cruz, Texas, USA) and CD68 (PGM1, 1:100, DAKO, Germany) was detected with Opal Multiplex 7-color manual kit (Akoya Bioscience, USA). Slides were scanned by the Vectra Polaris ™ imaging system (Akoya Bioscience, USA) and analysed by InForm software (Akoya Bioscience) and HALO<sup>R</sup> (Indica Labs, United Kingdoms).

For quantification of cell types and assessment of the architecture 29 adrenal glands were evaluated (19 patients positive for SARS-CoV-2 and ten patients with influenza). Ten regions of interest (ROIs; 931 µm × 698 µm) of each adrenal gland were evaluated and the number of each cell type was quantified. For quantification of apoptosis serial sections of adrenal glands with inflammatory infiltrates ($n = 19$) was performed and ten regions of interest (ROIs; 931 µm × 698 µm) of each infiltrate were evaluated.

Histopathological evaluation and analysis of multispectral images was carried out independently by two experienced pathologists (MR, HB). For statistical analysis SPSS (IBM, New York, USA) was used. Comparison of numeric variables were conducted with Student´s $t$-test and for categorical variables, Fisher exact test was applied.

**Immunohistochemical detection and in situ hybridization of SARS-CoV-2 protein and RNA**. Antibodies and probes for detecting SARS-CoV-2 spike protein (s-SARS-CoV-2-s protein) in formalin-fixed paraffin-embedded (FFPE) tissue were first validated on SARS-CoV-2 infected and non-infected CACO2 cells (ATCC, HTB-37) that were processed to FFPE blocks (supplementary Fig. 1). For immunohistochemistry of s-SARS-CoV-2-s protein, the SARS-CoV-2-s antibody (clone 1A9, Genetex, USA) was used at a dilution of 1:80. Detection was carried out with MACH 3 Mouse HRP Polymer detection (Biocare Medical, USA) following the manufacturer´s protocol. For in situ hybridization of SARS-CoV-2 a sense probe (RNAscope CoV2019-S-sense, ACD Bio-Techne, USA) and an antisense probe (RNAscope CoV2019-S-antisense, ACD Bio-Techne) was used. Detection was carried out with Opal 570 (Akoya Bioscience) according to the manufacturer´s protocol. All slides were evaluated independently by two experienced pathologists (MR, HB).

**Cell culture experiments and western blot analysis**
*Cell culture*. SW13 cells (ATCC, CCL-105), HAC15 cells (ATCC, CRL-3301), CACO2 cells (ATCC; HTB-37), and Vero-E6 cells (ATCC, CRL-1586) were cultivated in culture medium ((SW13: L15, 10% fetal calf serum (FCS)), (HAC15:

DMEM, F12, ITS, 10% FCS), (CACO2 and VERO-E6 DMEM, 10% FCS)). Cells were routinely passaged when reaching a confluence of 80–90%.

*Isolation and expansion of a SARS-CoV-2 clinical isolate.* CACO2 cells cultivated in "virus isolation medium" (DMEM, 2% FCS, 100 U/mL penicillin-streptomycin, NEAA, 0.5 µg/mL gentamicin, and 0.25 µg/mL amphotericin B) were challenged for 2 h with a clinical isolate ((EU1, Pangolin lineage B.1.177, GISAID EPI ISL: 466888) previously obtained from a nasopharyngeal swab of a COVID-19 patient. Subsequently, the virus isolation medium was replaced with a regular culture medium, and three days postinfection supernatant was collected and passaged onto Vero-E6 cells in the virus isolation medium. After three additional days, cell culture supernatants were harvested and stored at −80 °C. Further expansion of viruses was performed in expansion medium (DMEM containing 5% FBS, 100 U/mL penicillin-streptomycin, NEAA). Virus stocks were characterized by RT-qPCR, as reported previously[34] the infectious titer of virus stock was $1.79 \times 10^5$ IU/ml (IU = infectious units), viral load of virus stock was $1.28 \times 10^{10}$ Geq/ml (Geq = genome equivalent). In parallel, near full-length genomes were generated from expanded stocks of SARS-CoV-2 following the ARTIC network nCoV-2019 sequencing protocol v2[35], as described previously[36].

*SARS-CoV-2 infection experiments.* For SARS-CoV-2 infection SW13 ($5 \times 10^4$ cells per well) and HAC15 cells ($5 \times 10^4$ cells per well) were plated in a 96-well plate (Sarstedt) in "virus infection medium" with low FCS ((SW13: L15, 5% FCS), (HAC15: DMEM, F12, ITS, 5% FCS)). Three independent experiments with each two technical replicates per condition were performed. Target cells were treated with either RDV (1 µM) 2 h before infection or were left untreated. Subsequently, cells were challenged with a serial dilution of a stock of an expanded SARS-CoV-2 clinical isolate (multiplicity of infection is given as supplementary table 1). Three hours postinfection, the infection medium was removed, cells were washed once with PBS and a fresh culture medium was added. A postwash sample was taken to determine the background of the residual input virus. Seventy-two hours post challenge, supernatants were lysed using the MagnaPure lysis buffer (MagNA Pure LC Total Nucleic Acid Isolation Kit—Lysis/Binding Buffer Refill; Roche). Cells were either detached and subsequently fixed for 90 min in 4% PFA/PBS (Applichem) or lysed in RIPA buffer (Cell Signaling Technology). All samples were heat-inactivated (65 °C for 15 min).

*Nucleic acid extraction and RT-qPCR for SARS-CoV-2 N1 gene.* Viral nucleic acid extraction of inactivated cell culture supernatants was done using the Beckmann Biomek NX robotics platform (Beckmann Coulter) and the RNAdvance Viral (Beckmann Coulter) according to the manufacturer's instructions. Subsequently, cDNA synthesis was performed using the High-Capacity RNA-to-cDNA (Thermo Fisher Scientific) according to the manufacturer's instructions. cDNA synthesis was performed for 60 min at 37 °C, 5 min at 95 °C on a PCR cycler (Eppendorf). RT-qPCR was performed using SARS-CoV-2 N1 gene primers (Qiagen, 222015; 500 nM) and probe (125 nM) (CDC) in a standard Taqman PCR in a QuantStudio 3 Real-Time PCR System (Thermo Fisher Scientific).

*Western blot analysis.* For western blot analysis cells were treated with cell lysis buffer (Laemmli buffer, Bio-rad) and heat-inactivated. Four COVID-19 patients with excellent tissue preservation were chosen to perform western blot analysis of adrenal tissue. For protein extraction, the Qproteome FFPE tissue kit (Qiagen, Hilden, Germany) was used following the manufacturer´s protocol. Thirty micrograms of protein were separated by sodium dodecyl sulfate polyacrylamide gel electrophoresis (SDS-PAGE, Invitrogen, Carlsbad, CA) and transferred to polyvinylidene fluoride membranes (0.45 µm, Immobilon Millipore, Bedford, MA). Nonspecific binding sites were blocked by incubation with 5% (wt/vol) nonfat dry milk in TTBS (0.1% Triton X-100, 20 mM Tris, 136 mM NaCl at pH 7.6) for 1 h. Subsequently, membranes were incubated with primary antibody (SARS-CoV-2 (Cell Signaling, USA), alpha-tubulin (Sigma-Aldrich, USA), TMPRSS2 (Bio SB, USA), ACE2 (Abcam, Germany), cleaved caspase 3 (Cell Signaling, USA), Il-6 (Cell Signaling, USA) (1:1000) for 12 to 14 h at 4 °C. After incubation for 1 h with HRP-conjugated secondary antibody (anti-mouse IgG (#7076) or anti-rabbit IgG (#7074) cell signaling technologies, 1:2000) immunoreactivity was visualized with Super-Signal West Pico Chemiluminescent kit (Pierce).

*MALDI-MSI experiments.* Tissue preparation steps for matrix-assisted-laser-desorption-ionization (MALDI) mass spectrometry imaging (MALDI-MSI) analysis was performed as previously described[37,38]. Although removal/reduced intensity of hydrophobic and unstable molecules during the deparaffinization process of FFPE samples were observed, there were still classes of robust metabolites not only chemically, but also spatially preserved in FFPE tissue specimens[39–42]. Recently, this protocol has been successfully applied in endocrine FFPE tissue to investigate the distribution of hormones and metabolites in the normal and diseased adrenal[43–45]. The reliability of the MS imaging protocol for FFPE tissues was demonstrated by a multicenter interlaboratory round-robin study which showed a high level of between-center reproducibility of FFPE tissue metabolite data[46].

FFPE sections were incubated at 60 °C for one hour and deparaffinized in xylene (2 × 8 min). The matrix solution consisted of 10 mg/ml 9-aminoacridine hydrochloride monohydrate (9-AA) (Sigma-Aldrich, Germany) in water/methanol 30:70 (v/v). SunCollect™ automatic sprayer (Sunchrom, Germany) was used for

matrix application. The MALDI-MSI measurement was performed on a Bruker Solarix 7 T Fourier-Transform-Ion cyclotron resonance-mass spectrometer (FT-ICR-MS) (Bruker Daltonik, Germany) in negative ion mode using 100 laser shots at a frequency of 1000 Hz. FT-ICR mass spectrometer (Bruker Daltonik) is equipped with a dual ESI-MALDI source and a SmartBeam-II Nd: YAG (355 nm) laser. The laser operated at a frequency of 1000 Hz using 100 laser shots. The laser energy value is 1.05 µJ/pulse. For the MSI experiments, the FT-ICR mass spectrometer (Bruker Daltonik, Bremen, Germany) equipped with a dual ESI-MALDI source was calibrated externally with L-arginine in the ESI mode and internally using the 9-AA matrix ion signal (m/z 193.077122) as lock mass. The mass spectra were root-mean-square normalized. The MALDI-MSI data were acquired over a mass range of m/z 75–1000 with 60 µm lateral resolution. Following the MALDI imaging experiments, tissue sections were stained with hematoxylin and eosin (H&E) and scanned with an AxioScan.Z1 digital slide scanner (Zeiss, Germany) equipped with a 20x magnification objective. After the MALDI-MSI measurement, the acquired data underwent spectra processing in FlexImaging v. 5.0 (Bruker Daltonics, Germany) and SCiLS Lab 20 (Bruker Daltonik GmbH).

Metabolite annotation was performed using the Human Metabolome Database (HMDB, http://www.hmdb.ca/)[47] and METASPACE (http://annotate.metaspace2020.eu/), which is a framework for false discovery rate (FDR)-controlled metabolite annotation at the level of the molecular sum formula for high-mass-resolution imaging mass spectrometry[48]. This annotation framework integrates verification processes considering metabolite-signal match (MSM) score by combining spectral and spatial measures, and decoy FDR-estimation approach with a decoy set generated via the use of implausible adducts. For identification of metabolites, MS/MS analysis was conducted using a continuous accumulation of selected ions mode, which allows the target ions to be selected in the quadrupole and accumulated in the collision cell. Quadrupole MS/MS were performed with an isolation width of 2 *m/z* and collision energy of 20 V with nitrogen as collision gas. Metabolites were identified by comparing the observed MS/MS spectra with standard compounds and/or by matching accurate mass with databases. Specifically, compounds with indications of being drug-, plant-, food-, or bacteria-specific were filtered stringently. Annotated endogenous metabolites based on HMDB, LipidMaps, and SwissLipid databases are given as source data files. For the identification of metabolites illustrated in Fig. 6, phosphatic acid, phosphatidylinositol, and inositol cyclic phosphate were identified by matching accurate mass with databases, because those masses are unique to the molecules by matching with databases. Ribose phosphate were identified by in situ MS/MS experiments comparing the observed MS/MS spectra with standard compounds (Supplementary Fig. 2). Hierarchical clustering analysis and pathway enrichment analysis was carried out with the MetaboAnalyst 4.0 (http://www.metaboanalyst.ca)[49,50]. For hierarchical clustering analysis peak lists with respective intensities were uploaded. A heatmap was created to visualize sample clustering based on peak intensities. Each colored cell on the map corresponds to an intensity value, with samples in rows and features in columns. Euclidean distance and Ward's method were applied for clustering analysis. For pathway enrichment analysis, algorithms including a hypergeometric test for over-representation analysis and relative-betweenness centrality for pathway topology analysis were selected. Homo sapiens (KEGG) was specified as pathway library (http://www.genome.jp/kegg/)[51]. The pathway view was generated according to the *p* values from the pathway enrichment analysis and pathway impact values from the pathway topology analysis. Quantification results of cortisone is based on the relative peak intensity of cortisone. An unpaired *t*-test was used for statistical analysis. The measurement of MSI were carried out within a batch using external calibration with L-arginine and internal lock mass calibration with 9-AA matrix ion signal (m/z 193.077122). The mass spectra were root-mean-square normalized.

**Reporting summary**. Further information on research design is available in the Nature Research Reporting Summary linked to this article.

## Data availability
All data generated or analysed during this study are included in this published article (and its supplementary information files). Source data are provided with this paper.

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

## Acknowledgements

We thank Andrea Sendelhofert and Anja Heier (Institute of Pathology, LMU) for excellent technical assistance. Funding (M.R.): BMBF (Bundesministerium für Bildung und Forschung) Defeat Pandemics. Deutsche Forschungsgemeinschaft (DFG, German Research Foundation) project number: 314061271-TRR 205; SFB 824 C04; 444776998 (WI 5359/2-1). The funder of the study had no role in study design, data collection, data analysis, data interpretation, or writing of the report.

## Author contributions

MR, HB, OTK, AW, and TK designed the study. TP and SL wrote the manuscript. BM, TP, SL, HB, NS, CD, VR, JH, SM, GP, YSK, TAW, JN, HS, MS, MG, FS, SS, RS, and TK performed experiments and collected or analyzed the data. All authors discussed the results. TK, MR, HB, and SM discussed and reviewed clinicopathological interpretations. All authors read, edited and approved the manuscript. The corresponding author had full access to all the data in the study and had final responsibility for the decision to submit for publication.

## Funding

## Competing interests
