## [Peer Review File · Nature Communications]

Reviewers' Comments:

Reviewer #1:

Remarks to the Author:

Comments for NCOMMS-21-01411

Comments to the Author:

In this manuscript, the authors found adrenal tropism of SARS-CoV-2 in critically ill patients accompanied with adrenalitis and pronounced damage of adrenal glands with disturbance of adrenal metabolism. A total of 31 adrenal gland autopsy specimens (*i.e.*, 21 patients who died from COVID-19 and 10 patients who died from pneumonia caused by influenza A virus H1N1) were used for histomorphological analysis, immunohistochemical evaluation, and MS-based metabolomic *in-situ* detection and imaging. The results clearly proven that the patients infected with the COVID-19 has a certain correlation with adrenal injury. These finds are very important for us to further understand the lethal mechanism of COVID-19. Although the paper is interesting, the authors still need to spend more time focusing on the following aspects to further improve the paper. My comments mainly focus on MALDI-MSI experiments. For immunohistochemical evaluation, it is out of my expertise.

Comments:

1. The English of this manuscript must be further polished and improved before resubmission to avoid grammar and spelling mistakes.
2. In MALDI-MSI experiments, the FFPE tissue sections were used for metabolomic *in-situ* detection and imaging. In order to ensure the ionization efficiency of metabolites, the deparaffinization is essential before MALDI matrix deposition, thus, the use of these organic solvents (such as xylene) and relative high temperature treatment (60 °C used in this paper) will inevitably result in loss, delocation, or even destruction of endogenous molecules in biological tissues. How does the author avoid or solve these issues? In order to obtain the reliable MS data, in my opinion, fresh tissue or tissue that has not been treated with any embedding medium would be a better choice for endogenous *in-situ* detection and imaging using a MALDI source MSI.
3. The authors claimed that a total of 4663 metabolite ion signals could be detected from the FFPE specimen tissue sections. These metabolite ion signals detected from which kinds of adrenal gland autopsy specimens? COVID-19 patients? Or influenza A virus H1N1 patients? Dose the number of the detectable metabolite ion signals is same or different? Detail information of these metabolite ion signals (such as *m/z*, intensity, identification, error/ppm, molecular formula, the name of identified compound, and so on) should be listed in Supplemental Materials.
4. In Figure 5B, only one mass spectrum was shown. The metabolite profiling MS data is inadequacy.
5. The authors did not clearly clarify how to identify these detected metabolite ion signals. If only by virtue of the high-resolution performance of the FTICR-MS (a Bruker Solarix 7T FT-ICR-MS used in this work) is not sufficient to accurately match the standard compounds in the database. *In-situ* detection MS/MS and *in-situ* tissue extraction-based LC-MS/MS data of the detectable metabolite ion signals are also needed for high accurate assignment and identification.
6. In MS experiments, how does the author perform external and internal calibration for mass spectral data acquisition? For data analysis, mass spectral data normalization processing is also missing in this manuscript. Detail information should be added in the Supplemental Information.
7. Why did the authors choose 9-AA as the MALDI matrix for *in-situ* detection of metabolite from adrenal gland autopsy tissue sections? Although 9-AA is an effective matrix for the negative ion mode detection of nucleosides, there are many other well-established matrices for comprehensive imaging of metabolites in tissues, such as 2-MBT, quercetin, DAN, DMCA, DPH, and so on. Therefore, the author needs to clarify the reasons for this choice.
8. Why did the author not try positive ion mode detection or both positive and negative ion mode detections of metabolites in tissues?
9. In addition, there is no information regarding experimental energy-per-pulse values (or simply laser attenuation values if the energy per-pulse of the laser cannot be measured experimentally) used for acquiring the MALDI spectra using 9-AA as a matrix. Without this information is difficult for me to judge the performance of the 9-AA matrix in terms of the energy thresholds required to produce primary and secondary ions.
10. Quantification results of cortisone is very important for this manuscript, therefore, the

evaluations of repeatability, reproducibility, intra-/inter-day stability, and linearity are essential in MS-based quantitative experiments.

11. Scale bar should be added in all optical images of H&E-stained tissue sections and selected metabolite ion images in Figure 5D.

12. The abbreviations shown in this paper should be given in full names at the first appearing place in text, such as "MALDI", "MSI", "FT-ICR", and so on.

13. The words of "visualise" and "deparaffinised" used in this paper should be changed to "visualize" and "deparaffinized", respectively.

Reviewer #2:

Remarks to the Author:

This is an interesting study providing additional data from what has already been published about patients with COVID-19 (C) and adrenal hemorrhage and/or adrenal dysfunction (AH/D), compared to patients with influenza (I). Although the data are interesting, I think for a high profile paper that supports the claims of the authors more proof is needed:

1. What are the comorbidities of these patients? could the latter be responsible for the difference, C vs. I?
2. There is a lot of inflammation and I wonder whether the virus simply finds its way to the SF1 expressing cells rather than directly infecting it (from the circulation). The only way to differentiate between the two is to use adrenal cell lines (multiple ones, infect with the viruses and compare A vs. I).
3. One of the most effective meds against C is dexamethasone. How do the authors explain the havoc here in the adrenal tissue, that is full of glucocorticoids? in vitro experiments with cell lines will need to address the mechanism
4. The fact that even few patients with I had similar adrenal issues, adds to the doubts over the main thesis of the paper. Ultimately, it is not clear to this reviewer that the adrenal zonation issues are due to the infection; they may rather due to inflammation, which as the authors point out is higher among I patients...

Reviewer #3:

Remarks to the Author:

Paul et al. analyzed whether adrenal glands express SARS-CoV-2 entry proteins ACE2 and TMRPSS2 and whether they are infected in COVID-19 deceased patients. The topic is of great interest but the data, as presented, are not convincing to be recommended for publication in Nat Com.

The manuscript needs to be improved in several ways. Introduction and discussion can be more structured and streamlined to put the findings and their physiological implications into context. Improved figure design and labeling as well as a higher number of representative images illustrating the observations will strengthen the study. The limitations of the study such as retrospective analysis after death and lack of laboratory infection data should be discussed.

Authors analyzed adrenal gland autopsy specimens from 21 patients, however, only a single biopsy was analyzed for ACE2 and TMRPSS2 expression and SARS-CoV-2 RNA. If all the biopsies are available, more samples should be analyzed for expression of SARS-CoV-2 entry cofactors and viral antigen and/or RNA, to get a more comprehensive picture of adrenal tropism of SARS-CoV-2. As stated above, it is difficult to understand the labeling of the figures, e.g in Fig. 1A -C, what is blue, what is brown, and has the anti-S antibody been evaluated for specificity against tissue derived from non-infected of Flu infected individuals).

Specific points:

- 52ff "Cytokine toxicity and autoimmunity can be reduced by a mild immunosuppression due to activation of the hypothalamus-pituitary-adrenal axis (HPA axis) with hypercortisolemia." Provide

reference

- Provide the exact name and catalogue number of the used antibodies. For all antibodies that have not been validated in this study please provide a reference or indicate that they have been validated by the manufacturer. In general, the antibody dilutions seem lower than usually applied.
- Generally, provide more images of more patients as supplemental figures. Provide at least one image per patient. If possible, quantify co-localisation of ACE2/TMPRSS2/Spike with markers for adrenal cortical cells/capillaries. For histological images, provide overview pictures with magnified inset images where possible
- 96 typo TMRPSS2
- 114 please show the validation data
- 161 Is this a usual finding that TMPRSS2 localizes in the nucleus and cytoplasm?
- Fig 1 Please confirm the findings (ACE2, TMPRSS2, Spike expression) by Western blot of tissue lysates if possible
- Fig 1C, no spike protein detection in capillaries?
- Fig 2C only one apoptotic cell? Is this representative? Provide more images and quantify how many inflammatory infiltrates contain apoptotic cells
- English decimal is indicated by point not comma, check spellings: spike vs S protein vs S-protein, Caco-2, Covid-19 vs COVID-19, inconsistent across the manuscript
- Fig 2H I K how was this counted? Of how many patients and images?
- Fig2 provide images of influenza sections
- In general provide more images with lower magnification which allows better evaluation of the distribution
- Order of figure labeling in legends and in text confusing and not in order of appearance. Please write figure legends so that it can be followed what has been done experimentally.
- No evidence of direct hypothalamic or pituitary effects is found. Please show representative figures. Please discuss, how infection of the adrenal gland might still impact the HPA axis. Discuss relevant publications indicating that such dysregulations have been observed in COVID-19 patients
- Fig 5A labeling not readable
- 301 have such antibodies reacting to ACTH been found in COVID-19 patients? Otherwise this statement sounds very speculative
- 305 how does hypercortisolemia fit to the findings in fig 5c
- 311 Always give references if statements are made
- 319 how is the HPA Axis monitored in the clinical setting? How should be responded dependent on what observations? Or is simply suggested to monitor the axis to gain new insights into the course and risk of severe disease?

Thank you for giving us the opportunity to revise our manuscript. We are excited to resubmit to you the revised version of manuscript titled “*Adrenal tropism of SARS-CoV-2 and adrenal findings in patients with severe fatal COVID-19: a post-mortem case series*” for publication in *Nature Communications*.

We would also like to take this opportunity to express our thanks to the reviewers for the positive feedback and constructive comments and suggestions, which we believe have resulted in an improved revised manuscript.

We have addressed the reviewers’ concerns with additional experiments, including multiple new images as figures and additional text in the manuscript

Reviewer 1:

1. *The English of this manuscript must be further polished and improved before resubmission to avoid grammar and spelling mistakes.*
The manuscript was edited by a professional editing service to avoid any mistakes.
2. *In MALDI-MSI experiments, the FFPE tissue sections were used for metabolomic in-situ detection and imaging. In order to ensure the ionization efficiency of metabolites, the deparaffinization is essential before MALDI matrix deposition, thus, the use of these organic solvents (such as xylene) and relative high temperature treatment (60 °C used in this paper) will inevitably result in loss, delocation, or even destruction of endogenous molecules in biological tissues. How does the author avoid or solve these issues? In order to obtain the reliable MS data, in my opinion, fresh tissue or tissue that has not been treated with any embedding medium would be a better choice for endogenous in-situ detection and imaging using a MALDI source MSI.*

We thank the reviewer for these very important comments. We agree with the reviewer that we would also favour fresh tissues for *in situ* metabolomics study. However, due to the biosafety regulations for handling infectious tissues, the samples available for this study are only formalin-fixed paraffin-embedded (FFPE) adrenal gland autopsy specimens.

The protocol of metabolite imaging of FFPE tissue applied in this study was published several years ago (PMID: 27414759, PMID: 25965788). Although removal/reduced intensity of hydrophobic and unstable molecules during the deparaffinization process of FFPE samples were observed, there were still classes of robust metabolites not only chemically, but also spatially preserved in FFPE tissue specimens (PMID: 27414759, PMID: 25965788). This protocol has been extensively used in distinct FFPE clinical tissues under physiological and pathophysiological conditions (e.g., PMID: 31148595, PMID: 30580496, PMID: 29035378, PMID: 28978092). Recently, this protocol has been successfully applied in endocrine FFPE tissue to investigate the distribution of hormones and metabolites in the normal and diseased adrenal (PMID: 31492715, 31484828, PMID: 31957522). The reliability of the MS imaging protocol for FFPE tissues was demonstrated by a multicentre interlaboratory round-robin study which showed a high level of between-centre reproducibility of FFPE tissue metabolite data (PMID: 29968108). In addition, there are multiple mass spectrometry studies, based on liquid- and gas-chromatography MS, demonstrating that metabolites are reliably retained in FFPE tissue samples (PMID: 21984915;

PMID: 22498707; PMID: 26348873; PMID: 26415588; PMID: 28074002; PMID: 31375752). A recently published protocol for metabolomic and lipidomic profiling in formalin-fixed paraffin-embedded kidney tissue by LC-MS with subsequent detection of selected lipid species by an independent in situ MS imaging approach demonstrates the complementary use of both techniques (PMID: 33059858).

Accordingly, we added the following text to the manuscript (page 13 and 14):

“Tissue preparation steps for matrix-assisted-laser-desorption-ionisation (MALDI) mass spectrometry imaging (MALDI-MSI) analysis was performed as previously described (PMID: 27414759, PMID: 25965788). Although removal/reduced intensity of hydrophobic and unstable molecules during the deparaffinization process of FFPE samples were observed, there were still classes of robust metabolites not only chemically, but also spatially preserved in FFPE tissue specimens. This protocol has been extensively used in FFPE clinical tissues under physiological and pathophysiological conditions (PMID: 31148595, PMID: 30580496, PMID: 29035378, PMID: 28978092). Recently, this protocol has been successfully applied in endocrine FFPE tissue to investigate the distribution of hormones and metabolites in the normal and diseased adrenal (PMID: 31492715, 31484828, PMID: 31957522). The reliability of the MS imaging protocol for FFPE tissues was demonstrated by a multicentre interlaboratory round-robin study which showed a high level of between-centre reproducibility of FFPE tissue metabolite data (PMID: 29968108).”

3. The authors claimed that a total of 4663 metabolite ion signals could be detected from the FFPE specimen tissue sections. These metabolite ion signals detected from which kinds of adrenal gland autopsy specimens? COVID-19 patients? Or influenza A virus H1N1 patients? Does the number of the detectable metabolite ion signals is same or different? Detail information of these metabolite ion signals (such as m/z, intensity, identification, error/ppm, molecular formula, the name of identified compound, and so on) should be listed in Supplemental Materials.

We thank the reviewer for the comment and suggestions.

4663 ion signals are the total number of MS peaks detected in all tissues, including COVID-19 and influenza adrenal glands. However, they are with different ion intensities in both groups. These 4663 MS peaks were then submitted to metabolite databases for annotation based on matching of the measured masses to the databases (Metaspace, HMDB, LipidMaps and SwissLipid databases) and followed pathway enrichment analysis to identify discriminative metabolic pathways in COVID-19 and influenza groups, respectively. Additionally, metabolite-signal match (MSM) score combining spectral and spatial measures, and decoy false discovery rate (FDR)-estimation approach with a decoy set generated via the use of implausible adducts were integrated as verification processes to validate evidence of accurate metabolite annotation (<http://annotate.metaspace2020.eu/>, PMID: 27842059). Specifically, compounds with indications of being drug-, plant-, food-, or bacteria-specific were filtered stringently. We added Supplementary Table 1, which includes annotated endogenous metabolites based on HMDB, LipidMaps and SwissLipid databases.

4. In Figure 5B, only one mass spectrum was shown. The metabolite profiling MS data is inadequacy.

Mass spectrum showed in Figure 5B is an average mass spectrum of all measured tissues, including COVID-19 adrenal gland and influenza adrenal gland. To illustrate the mass spectrum more clearly, we modified Figure 5B (new Figure 6B) to include both the average mass spectrum of COVID-19 adrenal gland (red) and influenza adrenal gland (green).

5. The authors did not clearly clarify how to identify these detected metabolite ion signals. If only by virtue of the high-resolution performance of the FTICR-MS (a Bruker Solarix 7T FT-ICR-MS used in this work) is not sufficient to accurately match the standard compounds in the database. In-situ detection MS/MS and in-situ tissue extraction-based LC-MS/MS data of the detectable metabolite ion signals are also needed for high accurate assignment and identification.

We thank the reviewer for the comment and agree that this is an important point.

This study is a non-targeted metabolomics study for comprehensive and systematic identification of a wide range of metabolites from different metabolic pathways. It is an unbiased metabolomics approach for discovery and hypotheses generation. As MALDI-FTICR-MS has a high-resolving power sub-ppm mass accuracy, it can resolve metabolite peaks with similar nominal masses of compounds, metabolites and endogenous species in full scan mode, often without the requirement for MS/MS. Additionally, metabolite-signal match (MSM) score by combining spectral and spatial measures and decoy FDR-estimation approach with a decoy set generated via the use of implausible adducts were integrated as verification processes for metabolite annotation (<http://annotate.metaspaces2020.eu/>, PMID: 27842059).

For further MS/MS-based metabolite validation of specific metabolites, due to the limited amount of material, we cannot perform tissue extraction for LC-MS/MS experiments. Instead, we performed tissue MS/MS fragmentation experiments by comparing the observed MS/MS spectra with standard compounds.

We have added the following phrase to the text (page 14, 15):

“Metabolite annotation was performed using the Human Metabolome Database (HMDB, <http://www.hmdb.ca/>) (PMID: 29140435) and METASPACE (<http://annotate.metaspaces2020eu/>), which is a framework for false discovery rate (FDR)-controlled metabolite annotation at the level

of the molecular sum formula for high-mass-resolution imaging mass spectrometry (PMID: 27842059). This annotation framework integrates verification processes considering metabolite-signal match (MSM) score by combining spectral and spatial measures and decoy FDR-estimation approach with a decoy set generated via the use of implausible adducts.

To identify metabolites, MS/MS analysis was conducted using the continuous accumulation of selected ions mode, which allows the target ions to be selected in the quadrupole and accumulated in the collision cell. Quadrupole MS/MS analyses were performed with an isolation width of 2 m/z and collision energy of 20 V with nitrogen as collision gas. Metabolites were identified by comparing the observed MS/MS spectra with standard compounds and/or matching accurate mass with databases. To identify the metabolites illustrated in Figure 6, phosphatic acid, phosphatidylinositol, and inositol cyclic phosphate were identified by matching accurate masses with databases because those masses are unique to the molecules. Ribose phosphate was identified by in-situ MS/MS experiments comparing the observed MS/MS spectra with standard compounds (Supplementary Figure 2)."

Supplementary Figure 2: Ribose phosphate was identified by in-situ MS/MS experiments comparing the observed MS/MS spectra (green) with standard compounds (red).

6. In MS experiments, how does the author perform external and internal calibration for mass spectral data acquisition? For data analysis, mass spectral data normalisation processing is also missing in this manuscript. Detail information should be added in the Supplemental Information.

We thank the reviewer for this comment. We added the following text to the Material and Methods part (page 14):

“For the MSI experiments, the FTICR mass spectrometer (Bruker Daltonik, Bremen, Germany) equipped with a dual ESI-MALDI source was calibrated externally with L-arginine in the ESI mode and internally using the 9-AA matrix ion signal (m/z 193.077122) as lock mass. The mass spectra were root-mean-square normalised.”

7. Why did the authors choose 9-AA as the MALDI matrix for in-situ detection of metabolite from adrenal gland autopsy tissue sections? Although 9-AA is an effective matrix for the negative ion mode detection of nucleosides, there are many other well-established matrices for comprehensive imaging of metabolites in tissues, such as 2-MBT, quercetin, DAN, DMCA, DPH, and so on. Therefore, the author needs to clarify the reasons for this choice.

We thank the reviewer for this important comment.

9-AA matrix has been demonstrated as an effective matrix for in situ metabolomics imaging on clinical FFPE tissue under physiological and pathophysiological conditions (PMID: 27414759, PMID: 25965788, PMID: 31148595, PMID: 30580496, PMID: 29035378, PMID: 28978092). Most recently, this protocol was successfully applied on endocrine tissues for hormone and metabolite identification in the normal and diseased adrenals (PMID: 31492715, 31484828. PMID: 31957522). Therefore, for this COVID-adrenal study, we chose this well-established protocol using 9AA as a matrix, which yielded great sensitivity of the metabolite analysis that is advantageous for the simultaneous detection of a variety of cellular metabolites (Vermillion-Salsbury, R. L.; Hercules, D. M. *Rapid Commun. Mass Spectrom.* 2002, 16, pages 1575-1581, PMID: 20014780, PMID: 21043438, PMID: 20408595). We agree with the reviewer’s opinion that other matrixes could also be applied. Nevertheless, we have the most experience with 9-AA matrix on FFPE tissue and therefore decided to use the 9AA matrix for this study.

8. Why did the author not try positive ion mode detection or both positive and negative ion mode detections of metabolites in tissues?

9-Aminoacridine (9AA) is preferentially used as a matrix for negative mode matrix-assisted laser desorption/ionisation (MALDI) because 9AA is a moderately strong base ($pK = 9.99$) and readily accepts protons leading to the formation of $[M-H]^-$ species. The mechanism by which 9AA brings about ionisation in the negative mode appears to involve abstracting a labile proton in an acid/base reaction (Vermillion-Salsbury, R. L.; Hercules, D. M. *Rapid Commun. Mass Spectrom.* 2002, 16, pages 1575-1581; PMID: 19690837).

9. In addition, there is no information regarding experimental energy-per-pulse values (or simply laser attenuation values if the energy per-pulse of the laser cannot be measured experimentally) used for acquiring the MALDI spectra using 9-AA as a matrix. Without this information is difficult for me to judge the performance of the 9-AA matrix in terms of the energy thresholds required to produce primary and secondary ions.

It is challenging to measure the energy per-pulse of the laser experimentally. Bruker Daltonik (Bremen, Germany) kindly supported the experimental determination of the energy-per-pulse value. Accordingly, we added the following text on page 7 to describe detailed laser settings:

“FTICR mass spectrometer (Bruker Daltonik) was equipped with a dual ESI-MALDI source and a SmartBeam-II Nd: YAG (355 nm) laser. The laser operated at a frequency of 1000 Hz using 100 laser shots. The laser energy value was 1,05 μ J/pulse.”

10. Quantification results of cortisone is very important for this manuscript, therefore, the evaluations of repeatability, reproducibility, intra-/inter-day stability, and linearity are essential in MS-based quantitative experiments.

We fully agree that reliability and reproducibility are essential for quantitative experiments.

The quantification of cortisone is semi-quantitative based on the relative peak intensity of cortisone on tissue. The MS imaging experiments were carried out within one measurement in one batch, which maintained stability and avoided technical variation. Moreover, external calibration with L-arginine and internal lock mass calibration and normalisation procedures were integrated, allowing reliable relative quantification. The reproducibility of the protocol itself has been approved by “High-mass-resolution MALDI mass spectrometry imaging of metabolites from formalin-fixed paraffin-embedded tissue” published in Nature Protocol (PMID: 27414759) and a multicentre interlaboratory round-robin study (PMID: 29968108).

Accordingly, we added the following text into the manuscript (page 15):

“The cortisone quantification results were based on the relative peak intensity of cortisone. An unpaired t-test was used for statistical analysis. The measurements of MSI were carried out within a batch using external calibration with L-arginine and internal lock mass calibration with 9-AA matrix ion signal (m/z 193.077122). The mass spectra were root-mean-square normalised.”

11. Scale bar should be added in all optical images of H&E-stained sections and selected metabolite ion images in Figure 5D.

In Figure 5D, scale bars were provided for selected H&E and metabolite ion images as the magnification was similar for all images displayed. For easier interpretation, we added scale bars for each H&E image and metabolite ion image, as suggested by the reviewer.

12. The abbreviations shown in this paper should be given in full names at the first appearing place in text, such as “MALDI”, “MSI”, “FT-ICR”, and so on.

We added full names for each abbreviation used at first appearance.

13. The words “visualise” and “deparaffinised” used in this paper should be changed to “visualize” and “deparaffinized”, respectively.

We used British English spelling and writing.

Reviewer 2:

1. What are the comorbidities of these patients? Could the latter be responsible for the difference, C vs. I?

We thank the reviewer for this valuable comment.

Comorbidities of our patients are given in Table 1.

The two groups had a mean age of 73 versus 76 years. The most prevalent comorbidities were hypertension (100% in COVID-19 patients versus 100% in Influenza patients) and cardiovascular disease (79% in COVID-19 patients versus 60% in Influenza patients; $p=0.5$), as well as diabetes mellitus (37% in COVID-19 patients versus 50% in Influenza patients; $p=0.37$). The observed comorbidities are in the spectrum of expected medical conditions of elderly patients. There is no significant difference between the two groups with respect to comorbidities. In our cohort, comorbidities could not explain the significant difference in adrenal pathology.

2. There is a lot of inflammation and I wonder whether the virus simply finds its way to the SF1 expressing cells rather than directly infecting it (from the circulation). The only way to differentiate between the two is to use adrenal cell lines (multiple ones, infect with the virus and compare A vs. I).

We thank the reviewer for the valuable comment and suggestions.

In fact, the virus will probably find its way to the adrenal cells from the circulation, and it might be difficult to differentiate true infection from just diffusion. We could demonstrate significant levels of SARS-CoV-2 spike protein and SARS-CoV-2 RNA in SF1 expressing cells (Figure 1). In addition, we could detect SARS-CoV-2 antisense RNA in adrenal cortical cells, which proves viral replication and is not compatible with pure diffusion. Moreover, viral infection was associated with apoptosis and inflammation in COVID-19 patients (new Figure 3). Cell damage shown by expression of cleaved Caspase 3 (Figure 3) was associated with low Cortisol-levels (Figure 6), which might enhance systemic and local hyperinflammation.

To strengthen our data on adrenal cortical tissue of COVID-19 patients, we followed the reviewer's suggestion and completed functional cell line experiments. Only a limited number of human adrenal cortical carcinoma cell lines are available (SW13, H295R with multiple substrains such as HAC15 (PMID 21924324)). For our experiments, we chose SW13 and HAC15 and not the parental cell line H295R, as the HAC15 clone is monoclonal and provides a more stable steroidogenic phenotype that produces cortisol (PMID 21924324). Cortisol production was measured in the medium by ELISA 24 hours after plating the cells. The expression of ACE2 and TMPRSS2 was measured by western blot prior to the infection experiments. HAC15 showed a significant expression of the two proteins, whereas SW13 showed only weak expression of ACE2 and no expression of TMPRSS2 and therefore was not likely to be infected.

We included the following methods and results in the manuscript (pages 4,5 and pages 12, 13, 14) and designed a new figure (Figure 2):

Cell culture

SW13 cells (ATCC), HAC15 cells (ATCC), CACO2 cells (ATCC) and Vero-E6 cells (ATCC) were cultivated in culture medium ((SW13: L15, 10% fetal calf serum (FCS)), (HAC15: DMEM, F12, ITS, 10% FCS), (CACO2 and VERO-E6 DMEM, 10% FCS)). Cells were routinely passaged when reaching a confluence of 80-90%.

Isolation and expansion of a SARS-CoV-2 clinical isolate

CACO2 cells cultivated in “virus isolation medium” (DMEM, 2% FCS, 100 U/mL penicillin-streptomycin, NEAA, 0.5 µg/mL gentamicin, and 0.25 µg/mL amphotericin B) were challenged for 2 h with a clinical isolate ((EU1, pangoline lineage B.1.177, GISAID EPI_ISL: 466888) previously obtained from a nasopharyngeal swab of a COVID-19 patient. Subsequently, the virus isolation medium was replaced with regular culture medium, and three days post infection supernatant was collected and passaged onto Vero-E6 cells in virus isolation medium. After three additional days, cell culture supernatants were harvested and stored at -80°C. Further expansion of viruses was performed in expansion medium (DMEM containing 5% FBS, 100 U/mL penicillin-streptomycin, NEAA). Virus stocks were characterised by RT-qPCR, as reported previously³³. In parallel, near full-length genomes were generated from expanded stocks of SARS-CoV-2 following the ARTIC network nCoV-2019 sequencing protocol v2³⁴, as described previously³⁵.

SARS-CoV-2 infection experiments

For SARS-CoV-2 infection SW13 (5 x 10⁴ cells per well) and HAC15 cells (5 x 10⁴ cells per well) were plated in a 96-well plate (Sarstedt) in “virus infection medium” with low FCS ((SW13: L15, 5% FCS), (HAC15: DMEM, F12, ITS, 5% FCS)). Two independent experiments with each two technical replicates per condition were performed. Target cells were treated with either RDV (1 µM) two hours before infection or were left untreated. Subsequently, cells were challenged with a serial dilution of a stock of an expanded SARS-CoV-2 clinical isolate. Three hours post infection, infection medium was removed, cells were washed once with PBS and fresh culture medium was added. A post wash sample was taken. 72 h post infection, supernatants were lysed using the MagnaPure lysis buffer (Magna Pure LC Total Nucleic Acid Isolation Kit - Lysis/Binding Buffer Refill; Roche). Cells were either detached and subsequently fixed for 90 min in 4 % PFA/PBS (Applichem) or lysed in RIPA buffer (Cell signalling technologies). All samples were heat inactivated (65 °C for 15 min).

Nucleic acid extraction and RT-qPCR for SARS-CoV-2 N1 gene

Viral nucleic acid extraction of inactivated cell culture supernatants was done using the Beckmann Biomek NX robotics platform (Beckmann Coulter) and the RNAdvance Viral (Beckmann Coulter) according to manufacturer’s instructions. Subsequently, cDNA synthesis was performed using the High-Capacity RNA-to-cDNA (Thermo Fisher Scientific) according to manufacturer’s instructions. cDNA synthesis was performed for 60 min at 37°C, 5 min at 95°C on a PCR cycler (Eppendorf). RT-qPCR was performed using SARS-CoV-2 N1 gene primers (500 nM) and probe (125 nM) (CDC) in a standard Taqman PCR in a QuantStudio 3 Real-Time PCR System (Thermo Fisher Scientific).

Western blot analysis

For western blot analysis, cell lysates were inactivated for 15 min at 65 °C. In addition, four COVID-19 patients with excellent tissue preservation were chosen to perform western blot analysis of adrenal tissue. The Qproteome FFPE tissue kit (Qiagen, Hilden, Germany) was used following the manufacturer's protocol for protein extraction. 30 µg of protein was separated by sodium dodecylsulfate polyacrylamide gel electrophoresis (SDS-PAGE, Invitrogen, Carlsbad, CA) and transferred to polyvinylidene-fluoride membranes (0.45 µm, Immobilon Millipore, Bedford, MA). Nonspecific binding sites were blocked by incubation with 5% (wt/vol) nonfat dry milk in TTBS (0.1% Triton X-100, 20 mM Tris, 136 mM NaCl at pH 7.6) for 1 hour. Subsequently, membranes were incubated with primary antibody (SARS-CoV-2 (Cell Signaling Technology, USA), alpha-tubulin (Sigma Aldrich, USA), TMPRSS2 (Bio SB, USA), ACE2 (Abcam, Germany), cleaved Caspase 3 (Cell Signaling Technology, USA), IL-6 (Cell Signaling Technology, USA) (1:1000)) for 12 to 14 hours at 4°C. After incubation for 1 hour with HRP-conjugated secondary antibody (Cell signaling Technology, 1:2000), immunoreactivity was visualised with SuperSignal West Pico Chemiluminescent kit (Pierce).

Viral infection and replication in adrenal cortical carcinoma cell line

To functionally validate our observations of autopsy tissue of COVID-19 patients, we performed infection experiments with two adrenal cortical cell lines (HAC15, which was positive for ACE2 and TMPRSS2 and SW13, which showed weak ACE2 expression and was negative for TMPRSS2) (Figure 2A). The antiviral drug remdesivir (RDV) served as infection control. Target cells were treated with either RDV (1 μ M) two hours before infection or were left untreated. Subsequently, cells were challenged with a serial dilution of a stock of an expanded SARS-CoV-2 clinical isolate. Viral replication of SW13 and HAC15 was assessed by RT-qPCR (SARS-CoV-2 N1 gene), both in a wash (postwash; time zero) and in the supernatant (harvest; time 72 hours). Additionally, cells were collected for histopathology and western blot analysis. SW13 showed no detectable virus release into the cell culture supernatant with background levels (postwash). However, virus replication with values clearly exceeding background levels was detectable in the supernatant of HAC15 (Figure 2B). Virus release could be blocked by RDV pretreatment. FISH analysis of cell blocks and western blot analysis confirmed these RT-qPCR results. In RDV treated HAC15, no sense or antisense SARS-CoV-2 RNA could be detected (Figure 2C). In contrast, untreated HAC15 showed virus infection and replication with positivity for antisense and sense SARS-CoV2 (Figure 2C) and expression of SARS-CoV2 protein (Figure 2D). Intriguingly, low viral replication was sufficient to induce apoptosis in HAC15 with cleavage of caspase 3 shown by western blot analysis (Figure 2D) and immunohistochemistry (Figure 2C). Moreover, SARS-CoV-2 triggered inflammatory response with upregulation of IL-6 (Figure 3D), which is elevated in COVID-19 patients and correlates with adverse clinical outcome (PMC7190990, PMC7080116).

Figure 2. Infection of adrenal cortical carcinoma cell lines with SARS-CoV-2 induces apoptosis and triggers an inflammation response

Western blot analysis (A) of SW13 and HAC15 shows moderate to high expression of ACE2 and TMPRSS2 in HAC-15 and low expression of ACE2 and no expression of TMPRSS2 in SW13; tubulin is shown as the loading control. (B) RT-qPCR analysis (N1 gene) of cell culture supernatants of infected SW13 and HAC15 (pretreated with RDV (red) or untreated (blue)). Error bars represent mean and standard deviation of two technical replicates from two independent experiments. In-situ hybridization and immunohistochemistry of cell blocks (C) of infected HAC15 (pretreated with RDV (RDV +) or untreated (RDV -); overview image (scale bar = 50 μ m) and magnified insets (scale bar = 10 μ m)). In untreated HAC15 antisense and sense RNA (red dots) and protein of cleaved caspase 3 is detected, whereas in pretreated HAC15 no RNA or cleaved caspase 3 is visible (nuclear

counterstain (DAPI, hematoxylin) in blue). Western blot analysis of HAC15 (control, infected (untreated (RDV -) and infected (pretreated with RDV (RDV +)) is shown in D. In infected untreated HAC15 SARS-CoV-2 protein is detected, expression of cleaved caspase 3 and Il-6 is upregulated. In pretreated HAC-15 no SARS-CoV-2, cleaved caspase 3 or upregulation of Il-6 is visible.

3. One of the most effective meds against C is dexamethasone. How do the authors explain the havoc here in the adrenal tissue, that is full of glucocorticoids? In vitro experiments with cell lines will need to address the mechanism.

In fact, the reviewer is right. Treatment with dexamethasone is associated with reduced mortality for critically ill patients with COVID-19, which underlines the importance of our findings.

Depending on the concentration in the blood, glucocorticoids can cause immunosuppression. In the primaeval stage, glucocorticoids decrease inflammatory cell exudation, phagocytosis, and capillary dilatation, whilst they can inhibit cytokine release in the severe inflammatory stage. Intracellular glucocorticoid production in adrenal cortical cells generally does not limit local inflammation. Several conditions (such as autoimmunopathies or tuberculosis) can induce inflammation associated with damage and adrenal insufficiency (PMID:27099224).

To our knowledge, there is no data that dexamethasone can inhibit viral infection of SARS-CoV-2. As pointed out in answer 2, we proved viral replication in adrenal cortical cells in autopsy specimens and could observe local inflammation and apoptosis. Moreover, we performed additional functional experiments. There is only a limited number of human adrenal cortical carcinoma cell lines available (PMID 21924324). We chose two different adrenal cortical carcinoma cell lines (HAC15 (a substrain of H295R) and SW13). We chose HAC15 and not the parental cell line H295R, as the HAC15 clone is monoclonal and provides a more stable steroidogenic phenotype that produces cortisol (PMID 21924324). Cortisol production and the expression of ACE2 and TMPRSS2 were measured by ELISA and western blot before the infection experiments. 70% confluent cell lines were cultured (in triplicate) for 24 hours, and cortisol levels of supernatants were measured with ELISA following the manufacturer's protocol (Cortisol ELISA, cat. No. DEH3388 Demeditec Diagnostics GmbH, Kiel, Germany) using a microplate reader (Biorad, Germany) at 450 nm. For HAC15 and SW13 Cortisol, 352 ± 5 ng/ml and 12 ± 5 ng/ml were measured, respectively. Details of cell line experiments are given under answer 2, HAC15 but not SW13 (which was negative for TMPRSS2) could be infected with SARS-CoV-2. As shown by immunohistochemistry and western blot analysis, infection induced direct cytopathic effects with enhanced apoptosis and cleavage of caspase 3 (new Figure 2) which supports our autopsy specimens data. Cortisol production of HAC15 could not protect from viral infection or cell death.

We added the functional experiments in the results section, including an additional figure (see answer 2).

4. The fact that even few patients with I had similar adrenal issues, adds to the doubts over the main thesis of the paper. Ultimately, it is not clear to this reviewer that the adrenal zonation issues are due to the infection, they may rather due to inflammation, which as the authors point out is higher among I patients.

The doubts of the reviewer are based on a misunderstanding " ...due to inflammation, which is higher among I patients" ...; this is not correct. Inflammation was significantly higher in COVID-19 patients compared to influenza patients and not vice versa.

In our study, we observed SARS-CoV-2 tropism to the adrenal glands (shown by two different methods, immunohistochemistry and RNA in situ hybridisation). Moreover, sense and antisense SARS-CoV-2 RNA were detected in adrenal cortical cells indicative of viral replication. Viral tropism was associated with local inflammation, which was significantly higher among COVID-19 patients compared to influenza patients (see Figure 3A-K).

We fully agree with the reviewer that the severe damage of adrenal glands in COVID-19 patients might be a combination of direct infection associated with cell death (shown by cleavage of caspase 3, Figure 3C) and pronounced inflammation, which is our main conclusion of the study. To confirm viral infection of adrenal glands with a third independent method, we performed western blot analysis of adrenal tissue of COVID-19 patients and could detect high levels of s-SARS-CoV-2 protein in the lysates (revised Figure 1 Q).

Figure 1 Q: Western blot analysis of adrenal cortical tissue of COVID-19 patients

We are thankful for the reviewer's suggestion to perform functional experiments (see answer 2). We could infect the adrenal cortical carcinoma cell line HAC15 with SARS CoV-2. The viral infection led to cytopathic cell death and triggered an inflammatory response, which strengthens our main thesis (see answer 2).

Reviewer 3:

We thank the reviewer for his valuable comments. We revised the figures and figure legends and added multiple new images, which gives the reader a deeper insight into characteristic findings about COVID-19. In Figures 1, 2, 3, 4, and 5, overview images are provided with magnified insets. In addition, a comprehensive overview of all adrenal glands is given in Supplementary Figure 3.

Specific points:

1. 52ff "Cytokine toxicity and autoimmunity can be reduced by a mild immunosuppression due to activation of the hypothalamus-pituitary-adrenal axis (HPA axis) with hypercortisolemia". Provide reference.

We thank the reviewer for the comment, and the reference (PMC3978367) was added.

2. Provide the exact name and catalogue number of the used antibodies. For all antibodies that have not been validated in this study, please provide a reference or indicate that they have been validated by the manufacturer. In general, the antibody dilutions seem lower than usually applied.

We thank the reviewer for this very important comment.

We completely agree that it is essential to deliver all the antibodies' information and critically validate the antibodies before performing and interpreting immunohistochemistry or multiplex fluorescence. For this reason, Nature provides a reporting summary with all relevant information. Before usage, all antibodies passed an isotype and system control, and the antibodies were validated (see reporting summary). The dilutions of antibodies depended on the detection system used. We used a system with no further amplification step. Immunohistochemistry was tested in advance with positive and negative control tissue before applying it to the target tissue.

3. Generally, provide more images of more patients as supplemental figures. Provide at least one image per patient. If possible, quantify colocalisation of ACE2/TMPRSS2/Spike with markers for adrenal cortical cells/capillaries. For histological images, provide overview pictures with magnified inset images where possible.

We can understand that the reviewer wants to see more images.

In the manuscript, we show representative images. We followed the reviewer's suggestion and added multiple images as supplemental figures. In Supplementary Figure 3, an overview H&Es of the adrenal gland with magnified insets are provided for each patient (n=31).

In addition, we revised Figures 1,2,3,4 and replaced single magnifications with overview images and magnified insets.

To quantify colocalisation of ACE2, TMPRSS2, and s-SARS-CoV-2 with CD34 or SF1, we performed double fluorescence of all patient samples with positive SARS-CoV-2 PCR test (n=19). CD34 and SF1 were detected with OPAL 690 (shown in red), and ACE2, TMPRSS2 or s-SARS-CoV2 were detected with OPAL 620 (shown in green) (Figure 1D-I). For quantification, ten regions of interest per patient were evaluated. ACE2 colocalised with CD34 (95% ±3% of CD34 positive cells were double-positive for CD34 and ACE2) and SF1 (87% ± 4% of SF1 positive cells were double positive for SF1 and ACE2). TMPRSS2 colocalised with SF1 (84% ± 4% of SF1 positive cells were double positive for SF1 and TMPRSS2). However, no colocalisation with CD34 could be observed (0% of CD34 positive cells were double-positive for CD34 and TMPRSS2). For s-SARS-CoV-2, we observed heterogeneous expression in adrenal glands with multiple foci and more

significant variability between different patient samples. We could not detect s-SARS-CoV-2 colocalisation in CD34 positive cells (0% of CD34 cells were double positive for s-SARS-CoV-2). However, s-SARS-CoV-2 colocalised with SF1 (32% ± 11% SF1 positive cells were positive for s-SARS-CoV-2).

Methods, results and images were added to the manuscript (pages 3, 4, 12):

Figure 1. Adrenal tropism of SARS-CoV-2.

Double immunofluorescences are displayed as overview (scale bar = 100 µm) microphotographs with magnified insets (scale bar corresponds = 15 µm). ACE2 (green) and CD34 (red) (D), TMPRSS2 (green) and CD34 (red) (E), s-SARS-CoV-2 (green) and CD34 (red) (F), ACE2 (green) and SF-1 (red) (G), TMPRSS2 (green) and SF-1 (red) (H), and s-SARS-CoV-2 (green) and SF-1 (red) (I). ACE2 colocalises with SF-1 and CD34, whereas TMPRSS2 and SARS-CoV-2 only localises to SF-1 but not to CD34.

Supplementary Figure 3: H&E of adrenal glands of all patients (C = COVID-19; I = influenza)

H&E images of all adrenal glands (21 COVID-19 patients (C1-21) and 10 influenza patients (I1-10)) are given as an overview (scale bar = 800 μ m) and magnified inset (scale bar = 50 μ m)

96 typo TMRPSS2

Typo was corrected to TMRPSS2

114 please show the validation data

Validation data is added as Supplementary Figure 1

Supplementary Figure 1. Immunohistochemistry of s-SARS-CoV-2 of cell blocks of CACO2 uninfected or infected with SARS-CoV-2 (scale bar = 10 μ m).

161 Is this a usual finding that TMPRSS2 localises in the nucleus and cytoplasm?

TMPRSS2 is a serine protease that cleaves and activates viral envelope glycoproteins. For detection of TMPRSS2, we used the mouse monoclonal antibody BSB136 (BioSB, USA). The antibody was validated by the company with human lung tissue. Before staining the target tissue, we tested the antibody with human renal tissue and performed isotype and system controls. The data is shown below.

Immunohistochemistry of TMPRSS2 in renal tissue with isotype control (scale bar = 100 μ m).

In the datasheet, optional nuclear localisation of the antibody is reported. Cell data (human protein atlas) reports the main localisation to be intracellular and membranous with additional location to the nucleoplasm.

This localisation data is reflected by our findings in the adrenal glands with predominant intracellular expression and only a few cells showing nuclear expression of TMPRSS2 (Figure 1B).

Figure 1, please confirm the findings (ACE2, TMPRSS2, Spike expression) by Western blot of tissue lysates if possible.

We thank the reviewer for the suggestion to add western blot analysis of adrenal glands to confirm our data with a third method (immunohistochemistry, in situ hybridization and western blot analysis).

Due to biosafety regulations for handling infectious tissues, the samples available for this study were only formalin-fixed paraffin-embedded (FFPE) adrenal gland autopsy specimens. We chose four COVID-19 patients with excellent tissue preservation and low autolysis. For protein extraction we used the Qproteome FFPE tissue kit (Qiagen, Hilden, Germany) and followed the

manufacturer's protocol. We could detect variable amounts of ACE2, TMPRSS2, and s-SARS-CoV-2 protein in all samples analysed. Western blot analysis was added to Figure 1 (Figure 1 Q).

Figure 1 Q: Western blot analysis of adrenal gland tissue of COVID-19 patients (A1-A4).

Fig 1C, no spike protein expression in capillaries?

By immunohistochemistry and by additionally performed double immunofluorescence, we detected no expression of s-SARS-CoV-2 in capillaries. In contrast, we occasionally observed sense s-SARS-CoV2 positivity by in situ hybridization in CD34 positive cells (Figure 1P). Therefore, we also quantified colocalisation of CD34 with s-SARS-CoV-2 sense and antisense RNA in our patient samples (n=19). Only in a small subset of CD34 positive cells s-SARS-CoV-2 sense RNA ($8\% \pm 3\%$ of CD34 positive cells were positive for s-SARS-CoV2 sense RNA), but no antisense s-SARS-CoV2 RNA was detected. The small percentage of sense s-SARS-Cov-2 positivity in capillaries might be due to the trafficking of virus RNA. However, there is no proof of virus replication in adrenal capillaries, which were negative for TMPRSS2 and antisense RNA.

Fig 2c only 1 apoptotic cell? Is this representative? Provide more images and quantify how many inflammatory infiltrates contain apoptotic cells.

We thank the reviewer for the question and suggestion.

In Figure 2C, we show cleaved caspase 3 staining of inflammatory infiltrates. To quantify apoptotic cells in inflammatory infiltrates, we made serial sections of adrenal glands of 19 COVID-19 patients. Ten regions of interest (ROIs; $931\mu\text{m} \times 698\mu\text{m}$) per infiltrate were evaluated. In 79% of inflammatory infiltrates, we could detect at least one apoptotic cell per infiltrate (mean = 3 ± 3).

As suggested, we revised Figure 2 (now Figure 3), displaying more inflammatory infiltrates with cleaved caspase 3 staining (3C). In addition, we added the method and the quantification data to the manuscript.

Figure 3. Inflammation of adrenal glands from patients who died of COVID-19 or influenza.

H&E sections of adrenal glands of COVID-19 patient (A) and influenza patient (B) revealed lymphohistocytic infiltrates in the adrenal cortex of COVID-19 patients (overview scale bar = 800 μm and magnified inset scale bar = 50 μm). Multiplex immunofluorescence (D-G, scale bar = 50 μm) shows that infiltrates consisted of CD4 (D, yellow), CD8 (E, green), and CD68 (F, red) positive cells (DAPI counterstain in blue). Quantification of cell types and statistical analysis showed a significantly higher number of CD4 (H), CD8 (I), and CD68 (K) positive cells in patients with COVID-19 compared to influenza (p values were calculated using Student's t-test). Single-cell apoptosis could be detected in the centre of the inflammatory infiltrates by immunohistochemistry of cleaved caspase 3 (C, scale bar corresponds to 10 μm).

English decimal is indicated by point not comma, check spellings spike vs S protein versus S-protein, Caco-2, Covid-19 vs COVID-19, inconsistent across the manuscript.

We checked on spelling throughout the manuscript and used point for decimal. We now consistently use s-SARS-CoV-2 for spike protein, CACO2 and COVID-19.

Fig 2HIK how was this counted? Of how many patients and images?

The methodology for quantification of cell types is provided in the methods section. We supplemented the exact numbers of patients and images, and the following paragraph was added (page 12):

To quantify cell types and assess the architecture, 29 adrenal glands were evaluated (19 patients positive for SARS-CoV-2 and 10 patients with influenza). Ten regions of interest (ROIs; 931 μm x 698 μm) of each adrenal gland were evaluated, and the number of each cell type was quantified.

Fig2 provide images of influenza sections.

We thank the reviewer for the suggestion.

We included a supplementary figure (Supplementary Figure 3) with one image of each patient (including all influenza patients). Moreover, a representative image of an influenza patient was added to Figure 2 (now Figure 3B) (see above).

In general, provide more images with lower magnification which allows better evaluation of the distribution.

We followed the reviewer's suggestion, which allows the reader to better understand the distribution of protein expression and inflammation.

We revised Figure 1, Figure 2, and Figure 3 and replaced single magnifications for overview images and magnified insets or overview images combined with magnified images.

Figure 1. Adrenal tropism of SARS-CoV-2.

Overview images (scale bar = 100 μ m) with magnified insets (scale bar = 15 μ m) of immunohistochemistry of ACE2 (A), TMPRSS2 (B), and s-SARS-CoV-2 (C) are shown. Adrenal cells and intervening capillaries express ACE2, and TMPRSS2. s-SARS-CoV-2 protein can be detected in adrenal cortical cells. (D-I) double immunofluorescence of ACE2, TMPRSS2 and s-SARS-CoV-2 with CD34 or SF-1 are displayed as overview (scale bar = 100 μ m) with magnified insets (scale bar corresponds = 15 μ m). ACE2 colocalises with SF-1 and CD34, whereas TMPRSS2 and SARS-CoV-2 only localise to SF-1 but not to CD34. In-situ hybridization of adrenal tissue with s-SARS-CoV2 sense probe (K, L in red) and antisense probe (M, N in red) with DAPI counterstain (blue) (K, M (overview (scale bar = 50 μ m) and corresponding magnification (L, N) (scale bar = 15 μ m)) confirm the

presence of s-SARS-CoV-2 RNA in adrenal tissue. Viral RNA (red dots) is detected in SF1 positive adrenal cells (green, O) and in CD34 positive endothelial cells (green, P); scale bar corresponds to 10 μm , DAPI counterstaining in blue. (Q) western blot analysis of adrenal tissue of four different patients (A1-A4) shows variable expressions of ACE2, TMPRSS2 and s-SARS-CoV-2 in adrenal tissue (tubulin is shown as loading control).

Figure 4. Microthrombi and focal haemorrhage in adrenal glands from COVID-19 patients. Hematoxylin-eosin staining (A), and anti-fibrin immunohistochemistry (B) (overview (scale bar = 100 μm) and magnified inset (scale bar = 10 μm) show expanded capillaries with fibrinous microthrombi. In a subset of patients, fresh focal haemorrhage could be observed (C), masson's trichrome staining (overview scale bar = 100 μm and magnified inset scale bar = 10 μm).

Order of figure labelling in legends and in text confusing and not in order of appearance. Please write figure legends so that it can be followed what has been done experimentally.

Figure legends were revised.

No evidence of direct hypothalamic or pituitary effects is found. Please show representative figures. Please discuss, how infection of the adrenal gland might still impact the HPA axis. Discuss relevant publications indicating that such dysregulations have been observed in COVID-19 patients.

Representative figures (H&E) of the pituitary gland (F) and hypothalamus (G) are added to Figure 5:

The following paragraph and relevant references were added to the discussion (page 10):

In contrast to our remarkable findings in adrenal glands, we could not observe significant structural abnormalities or inflammatory infiltrates in the hypothalamic or pituitary glands of COVID-19 patients. However, the severe damage of adrenal glands might lead to primary adrenal insufficiency with dysregulation of the HPA axis and hypocortisolemia in the end. Adrenal insufficiency is not a rare manifestation in COVID-19. In addition to several case reports^{37,38 10,39-41}, some observational studies^{12, 42} also reported this presentation.

Fig5A labelling not readable

Labelling of Figure 5A was enlarged to make it readable.

301 have such antibodies reacting to ACTH been found in COVID-19 patients? Otherwise, this statement sounds very speculative

Like SARS-CoV, SARS-CoV-2 contains various amino acid sequences with homology to human adrenocorticotrophic hormone (ACTH) key residues and in SARS-CoV antibodies targeting ACTH could be found. To our knowledge, so far in COVID-19, it was not looked for such antibodies. We agree that our remark is speculative. However, we do not make a strong statement but discuss this as one possibility for adrenal insufficiency.

305 how does hypercortisolemia fit to the findings in fig 5C?

In acute stress situations, cortisol levels are influenced not only by secretion. Low serum protein binding and suppression of cortisol degradation can lead to hypercortisolemia. In the acute phase of COVID-19, hypercortisolemia can be observed. However, in severe longstanding infections and post SARS-CoV, hypocortisolemia is a characteristic finding (PMID: 33488517,

PMID: 16060914). Our findings are in line with these observations where adrenalitis and adrenal damage are associated with low cortisol levels measured in the tissue.

311 always give references if statements are made

The following reference was added:

Michelen M. *et al.* Characterising long COVID: a living systematic review *BMJ Glob Health* 6, 005427 (2021).

319 how is the HPA axis monitored in the clinical setting? How should be responded dependent on what observations? Or is simply suggested to monitor the axis to gain new insights into the course and risk of severe disease?

To our knowledge, the HPA axis is not monitored routinely so far in COVID-19 patients. In our opinion, it is far too early to propose definitive therapeutic guidelines. However, due to our findings and in light of repeatedly reported adrenal insufficiency in COVID-19 patients, we would recommend monitoring the HPA axis to gain new insights into the course and risk of severe disease.

Reviewers' Comments:

Reviewer #1:

Remarks to the Author:

The authors have performed additional experiments and have addressed all my previous comments satisfactorily. I believe this revised manuscript is ready for publication. It is really a very interesting and important study for the evaluation of SARS-CoV-2 and COVID-19 infection.

Reviewer #2:

Remarks to the Author:

The revised manuscript addressed many of the concerns of this reviewer. I do not understand the SW13 data (why it did not infect); have the authors checked the origin of their cells?

Reviewer #3:

Remarks to the Author:

Authors have addressed almost all concerns I had. The paper now contains sufficient data to support the authors conclusions of an adrenal tropism of SARS-CoV-2 with associated inflammation. I recommend to publish.

Thank you for giving us the opportunity to revise our manuscript. We are excited to resubmit to you the revised version of manuscript titled “*Adrenal tropism of SARS-CoV-2 and adrenal findings in patients with severe fatal COVID-19: a post-mortem case series*” for publication in *Nature Communications*.

We have addressed the reviewer’s concern as follows:

Reviewer 3:

The reviewer notes that data are not sufficient to show productive infection, and therefore we request that you remove claims of productive infection throughout the manuscript and discuss these data as preliminary evidence. Please also note to include all the additional experimental details discussed with you by e-mail in the manuscript, and include at least 3 biological replicates for the PCR experiments.

We eliminated claims of productive infection and discussed the data as preliminary. In addition, we added all details as revised materials and methods, results and supplementary table 1. We also included a third biological replicate for the PCR experiments.